# The E3 ubiquitin ligase WWP2 regulates pro-fibrogenic monocyte infiltration and activity in heart fibrosis

Huimei Chen [1,2,6] ✉, Gabriel Chew[1,6], Nithya Devapragash[1], Jui Zhi Loh[1], Kevin Y. Huang[1], Jing Guo[1], Shiyang Liu[1], Elisabeth Li Sa Tan[1], Shuang Chen[2,3], Nicole Gui Zhen Tee[4], Masum M. Mia[1], Manvendra K. Singh [1], Aihua Zhang[3], Jacques Behmoaras [1,5] ✉ & Enrico Petretto [1,2] ✉

Non-ischemic cardiomyopathy (NICM) can cause left ventricular dysfunction through interstitial fibrosis, which corresponds to the failure of cardiac tissue remodeling. Recent evidence implicates monocytes/macrophages in the etiopathology of cardiac fibrosis, but giving their heterogeneity and the antagonizing roles of macrophage subtypes in fibrosis, targeting these cells has been challenging. Here we focus on WWP2, an E3 ubiquitin ligase that acts as a positive genetic regulator of human and murine cardiac fibrosis, and show that myeloid specific deletion of WWP2 reduces cardiac fibrosis in hypertension-induced NICM. By using single cell RNA sequencing analysis of immune cells in the same model, we establish the functional heterogeneity of macrophages and define an early pro-fibrogenic phase of NICM that is driven by Ccl5-expressing Ly6c[high] monocytes. Among cardiac macrophage subtypes, WWP2 dysfunction primarily affects Ly6c[high] monocytes via modulating Ccl5, and consequentially macrophage infiltration and activation, which contributes to reduced myofibroblast trans-differentiation. WWP2 interacts with transcription factor IRF7, promoting its non-degradative mono-ubiquitination, nuclear translocation and transcriptional activity, leading to upregulation of Ccl5 at transcriptional level. We identify a pro-fibrogenic macrophage subtype in non-ischemic cardiomyopathy, and demonstrate that WWP2 is a key regulator of IRF7-mediated Ccl5/Ly6c[high] monocyte axis in heart fibrosis.

Chronic inflammatory diseases often progress to a fibrotic stage, the endpoint of dysfunctional tissue repair responses[1]. Fibrosis account for a significant proportion of the annual death rate worldwide and include myocardial fibrosis, a global health issue related to nearly all forms of heart disease[2]. Generally, myocardial inflammation precedes fibrosis characterized by abnormal accumulation of extracellular matrix (ECM), leading to the enlargement of the interstitium (hypertrophy), ultimately causing heart failure[3-6].

A large body of evidence points toward cell-to-cell cooperation driving the pathogenesis of fibrosis[1,7]. In the heart, local tissue fibroblasts, a highly heterogenous cell population that normally ensure tissue homeostasis, respond to tissue injury, and contribute to fibrosis

[1]Programme in Cardiovascular and Metabolic Disorders, Duke-NUS Medical School, 8 College Road, 169857 Singapore, Singapore. [2]Institute for Big Data and Artificial Intelligence in Medicine, School of Science, China Pharmaceutical University, Nanjing 210009, China. [3]Department of Nephrology, Children's Hospital of Nanjing Medical University, Nanjing 210008, China. [4]National Heart Centre Singapore, Singapore 169609, Singapore. [5]Centre for Inflammatory Disease, Imperial College London, Hammersmith Hospital, London W12 0NN, UK. [6]These authors contributed equally: Huimei Chen, Gabriel Chew. ✉e-mail: huimei.chen@duke-nus.edu.sg; jacquesb@duke-nus.edu.sg; enrico.petretto@duke-nus.edu.sg

by uncontrolled production of ECM[8]. Likewise, cells of the immune system contribute to cardiac development, composition, and function but can also cause irreversible tissue damage during progressive fibrosis[9,10]. Among those, monocytes/macrophages are critically involved in the regulation of fibroblast activation and fibrogenic responses in heart disease[11–13]. Cardiac macrophages belong to a heterogenous cell population that is normally maintained through local proliferation; they can expand in response to injury through the recruitment of circulating monocytes from blood[14]. Macrophage-derived factors (e.g., TGFβ) are critical drivers of fibroblast trans-differentiation into active cells (i.e., myofibroblasts) in the heart and other tissues[15,16], and monocyte/macrophage depletion strategies regulate the outcome of the cardiac healing[15,17,18]. Thus, macrophage-fibroblast crosstalk through either soluble growth factors, cytokines/chemokines, matrix metalloproteinases, miRNAs, small extracellular vesicles or through cell–cell interaction, operates in the heart and contributes to the regulation of fibrosis[19–27]. The availability of the cardiac cell atlas at a single-cell resolution further unraveled the molecular basis of cell-to-cell interactions, especially the macrophage-fibroblast mRNA networks during homeostasis and disease[28–30]. These studies brought definitive proof of the heterogeneity of cardiac macrophages[14,18,31–33] and further confirmed that specific macrophage subsets are implicated in the regulation of tissue fibrosis[18,34].

Among cardiac conditions that lead to myocardial fibrosis, non-ischemic cardiomyopathy (NICM) can cause left ventricular dysfunction whereby mid-wall fibrosis is a major risk of sudden cardiac death[35,36]. Hypertension is the leading cause of NICM, and recent studies suggest that macrophage-driven inflammation plays a critical role in the pathogenesis of NICM[17,37,38]. However, the ontogeny, heterogeneity, and subsequent regulatory function of macrophages during hypertension-induced NICM and fibrosis remain to be fully elucidated. We previously reported that the E3 ligase WWP2 regulates cardiac fibrosis and cardiac dysfunction following angiotensin II (Ang-II) treatment through activation of an "ECM gene network" in myofibroblasts[39]. In humans, a genetic variant within the *WWP2* locus coordinates a profibrotic ECM transcriptional program by modulating the expression of the ECM gene network in the diseased heart[39]. In addition to the master-regulatory effect of WWP2 on fibroblast activation, our results also suggested a macrophage-driven effect of WWP2 that remained to be explored[39].

Here we establish a functional novel role of WWP2 in cardiac Ly6c[high] monocytes, the main source of macrophage expansion during the early phase of heart fibrosis. WWP2 regulates the activation of these pro-inflammatory and pro-fibrogenic cardiac Ly6c[high] monocytes and macrophages, which control myofibroblast *trans*-differentiation. WWP2 dysfunction in the myeloid cell compartment is sufficient to ameliorate cardiac fibrosis and improve heart function in vivo.

## Results

### Single-cell RNA sequencing reveals the cardiac macrophage heterogeneity during NICM

To investigate the regulation of resident macrophages during NICM, we first assessed the number of macrophages in the left ventricle (LV) tissue of B6J mice following Ang-II infusion (500 ng/kg/min), a well-established non-ischemic model of arterial hypertension, cardiac hypertrophy, and fibrosis[40]. We measured cardiac macrophage numbers (CD45[+]CD64[+]CD11b[+]F4/80[+]Ly6G[–] cells[41,42], Supplementary Fig. 1a) during a time-course of cardiac injury. Ang-II-infusion increased the accumulation of macrophages in murine hearts with a peak at 7 days (Fig. 1a). This finding is consistent with the dynamics of cardiac macrophage accumulation following transverse aortic constriction (TAC)[41,43], another NICM model of pressure overload-induced hypertrophy and heart failure[44]. The macrophage peak at day 7 suggested that this time point is functionally implicated in the

progression of NICM. In fact, 7 days post-Ang-II administration, coincides with fibrogenic responses such as early extracellular matrix organization in cardiac tissue[45,46]. Hence, we reasoned that 7 days post-injury represents a critical time point for mediating macrophage-dependent cardiac injury, and that in-depth analysis of macrophages during homeostasis and NICM will be critical for downstream mechanistic studies of cardiac fibrosis.

To test the previous hypothesis, we performed single-cell RNA sequencing (scRNA-seq) in LV tissues post 7-day Ang-II administration and compared the results with saline-infused controls (Supplementary Fig. 1b). After standard data processing and quality control, scRNA-seq profiles of 5497 CD45[+] cells were obtained and further annotated into 6 main cell types, including T cells (T), B cells (B), Natural Killer cells (NK), granulocytes (GC), dendritic cells (DC), and macrophages (MΦ) (Fig. 1b and Supplementary Fig. 2a). Seurat's Louvain clustering algorithm[47] further identified seven discrete cell subpopulations within cardiac macrophages (Fig. 1c, and Supplementary Fig. 2b–e). These represent a diverse range of macrophage states of activation, such as 'homeostasis/repair', 'type I interferon-stimulated genes (ISGs)-related' and 'Ly6c[high] monocytes' (Fig. 1d, e). The cell clusters identified in this study mostly overlap with those identified in previously published murine cardiac transcriptional analyses[33,38]. For example, the 'homeostasis/repair' cluster identified in this study showed high similarity to the homeostatic Timd4 cluster[33]. Similarly, our 'Ly6c[high,' and 'Ly6c[low]' monocyte clusters were functionally conserved with the 'monocyte' cluster identified by Dick et al[33]. (Supplementary Fig. 3a). The recently reported transcriptomic signature of resident cardiac macrophages following Ang-II-infusion[38] showed the highest relative expression in our 'homeostasis/repair' cluster (Supplementary Fig. 3b). Thus, our cardiac macrophage clusters 2 and 4 show a pro-inflammatory state of activation, while clusters 0 and 1 are enriched for homeostasis/repair functions following 7-day Ang-II-infusion.

At baseline, the majority of cardiac macrophages belong to the 'homeostasis/repair' (43%, cluster 0) and 'AP-1' (33%, cluster 1) clusters. These baseline macrophages are characterized by the absence of 'Ly6c[high] monocyte' and by only 2% of 'ISGs-related' clusters (Fig. 1f). Ang-II-infusion causes a shift toward pro-inflammatory macrophage clusters (cluster 2 and 4), while it reduces the relative abundance of 'homeostasis/repair' and 'AP-1' macrophages. To infer the potential developmental relationships between the cardiac macrophage clusters during NICM, we reconstructed the pseudotime dynamics of the macrophage subpopulations using the Monocle algorithm[48] (Fig. 1g). The inferred trajectory starts from the 'Ly6c[low] monocytes' and progresses towards 'Ly6c[high] monocytes' (cluster 4) and 'ISGs-related' (cluster 2). In order to explore the existence of a potential trajectory between inflammatory (C4) and homeostatic (C0) clusters, we used Slingshot projection[49,50]. This showed a pseudotime trajectory connecting the 5 clusters (Fig. 1h), and linking C4 (Ly6c[high] monocytes) and C0 (homeostasis/repair) clusters. We confirmed the latter by using UMAP of Monocle trajectory (Supplementary Fig. 3c). These findings demonstrate the relationship between cluster 0 and cluster 4, suggesting that the increase in the number of cardiac macrophages following 7 days Ang-II-infusion (Fig. 1a) is most likely due to Ly6c[high] monocyte infiltration.

A specific gene transcript that is used to measure cardiac macrophage polarization is *CCR2*, a marker for infiltrated monocyte-derived macrophages in the heart[17]. Ccr2[high] macrophages are pro-inflammatory, while Ccr2[low] macrophages correspond to a reparatory cell phenotype[31]. Accordingly, the 'ISGs-related' (cluster 2) and 'Ly6c[high] monocyte' (cluster 4) clusters contain relatively increased *Ccr2*-expressing cells when compared with the 'homeostasis/repair' (cluster 0) and 'AP-1' (cluster 1) clusters (Supplementary Fig. 3d). Furthermore, in dilated cardiomyopathy (DCM), the high and low *Ccr2* expressing clusters functionally correlate with *Ccr2*+ and *Ccr2*− macrophage subsets, respectively[51] (Supplementary Fig. 3e). A recent study

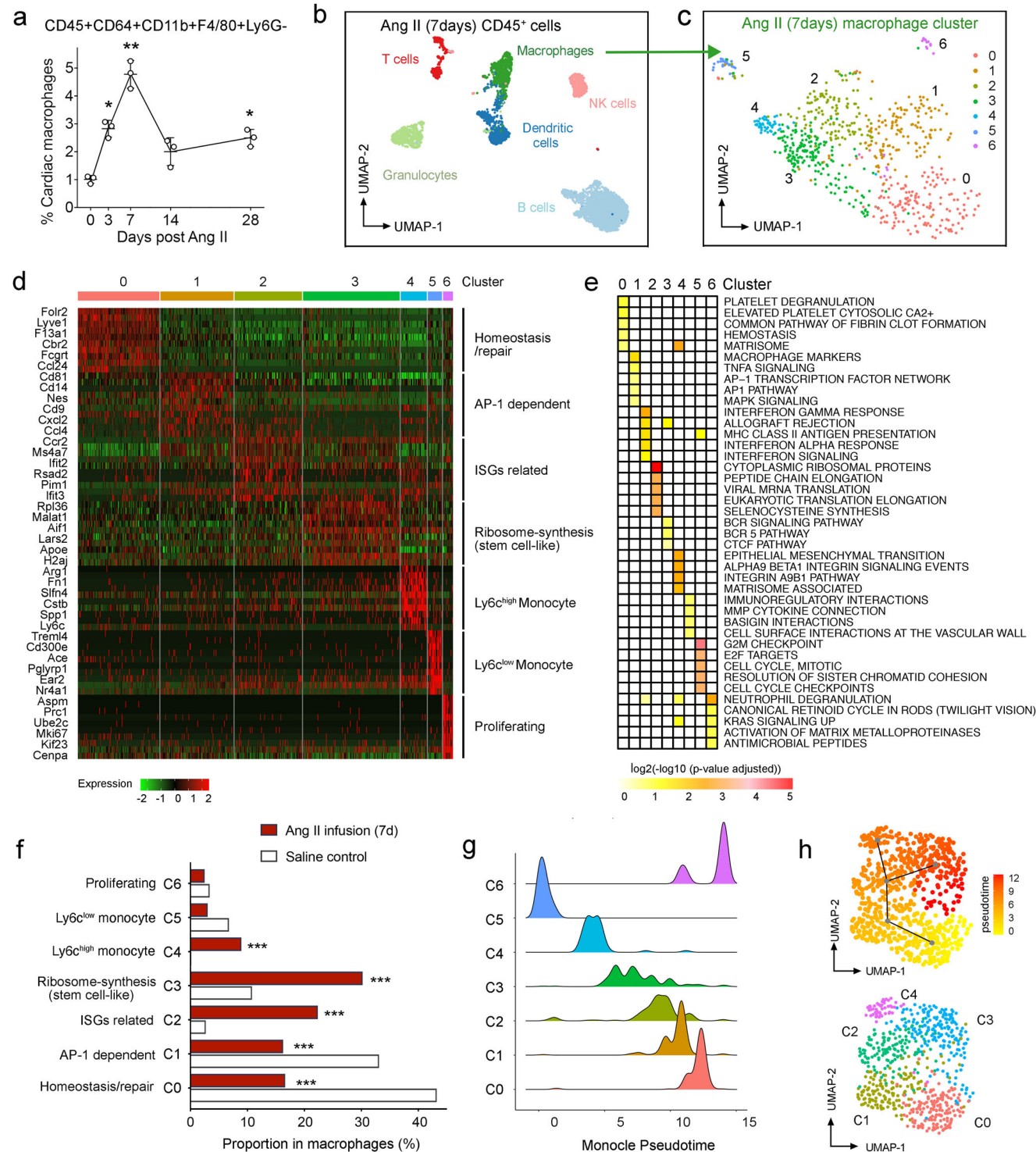

**Fig. 1 | Single-cell transcriptomic profiling and trajectory analysis of cardiac macrophages in a murine model of NICM. a** Dynamics of cardiac macrophages (live CD45+ CD64+ CD11b+ F4/80+ Ly6G−) from B6J mice treated with Ang-II (500 ng/kg/min). Cardiac macrophages are shown as a percentage with respect to total number of cardiac cells. Three biological replicates were used at each time point. P-values calculated by Mann–Whitney U test, vs percentage at day 0. *P < 0.05, **P < 0.01. **b** Cardiac CD45+ cells sorted from left ventricle (LV) from B6J mice treated with Ang-II (500 ng/kg/min, 7 days) and saline (controls) were pooled for scRNA-seq analysis on 5497 cells. Global uniform manifold approximation and projection (UMAP) dimension reduction analysis identified six major cell types. DC dendritic cells, NK cells natural killer cells. **c** Macrophages were re-clustered using Seurat's Louvain clustering. **d** Heatmap of the scaled gene expression of top 10 markers for each macrophage cluster from Fig. 1c, based on average log₂ fold change (FC). ISGs Interferon (IFN)-stimulated genes, AP-1 Activator protein 1. **e** Heatmap of enriched pathways identified in each cardiac macrophage cluster. The top five terms are shown, and the significance of enrichment is represented by log₂(−log₁₀ (adjusted p value)). **f** Bar plot showing the relative proportions of macrophage clusters from **c** in B6J control (saline) and B6J mice treated with Ang-II (500 ng/kg/min, 7 days). A two-proportion z test was used to assess statistical significance; ***P < 0.001 Ang-II-treated vs saline-treated control mice. **g** Monocle-derived pseudotime cell trajectory of each cardiac macrophage cluster from **c**. **h** UMAP of cardiac macrophages describing developmental trajectories superimposed (upper) on clusters from C0 to C4 (lower). The main trajectories are generated by Slingshot, which are calculated as the average pseudotime over all cells.

identified CD72 as a marker for infiltrating monocytes/macrophages in the TAC-induced hypertensive hearts[52]. We found that the expression of *Cd72* and *Ccr2* had significant similarity along the pseudotime trajectory in cardiac macrophages (Supplementary Fig. 3f). In addition, the expression of *Cd72* was highest in 'ISGs-related' clusters (cluster 2, Supplementary Fig. 3g), confirming its association with pro-inflammatory Ccr2[high] monocytes/macrophages in the heart.

In summary, cardiac macrophages from Ang-II-treated mice derive mainly from circulating Ly6c[high] monocytes (which present as Ccr2[high] cells), while the relative proportions of Ccr2[low] 'homeostasis/repair' and 'AP-1' macrophages diminish.

## WWP2 regulates cardiac monocyte infiltration and macrophage expansion through Ccl5

We previously showed that in Ang-II-induced NICM, WWP2 lacking the N-terminal region (WWP2[Mut/Mut]) exerts a cardio-protective function by regulating cardiac myofibroblasts with a contribution from immune cells[39]. However, the relative role and contribution of WWP2 in immune *vs* stromal cells remained to be elucidated. Given the importance of macrophage *WWP2* in pro-inflammatory macrophage polarization[53] and the significance of monocyte/macrophage contribution to cardiac fibrosis[13], we hypothesized that WWP2 modulates cardiac macrophage function at early phase of fibrosis in NICM. The single-cell transcriptional activation of cardiac macrophages identified earlier (Fig. 1) provided the experimental platform to test this hypothesis. We thus performed scRNA-seq in cardiac CD45[+] cells from wild-type (WT) and WWP2[Mut/Mut] mice subjected to 7 days of Ang-II treatment or saline (Fig. 2a). The *Wwp2* gene is evenly expressed within the WT cardiac macrophage populations (Supplementary Fig. 4a).

We report significant differences in the proportion of macrophage subpopulations between WT and WWP2[Mut/Mut] mice following Ang-II treatment, including 'homeostasis/repair' macrophages (cluster 0), 'stem-cell-like' macrophages (cluster 3), and Ly6c[high] monocytes (cluster 4) (Fig. 2b). The relative proportion of 'homeostasis/repair' macrophages (cluster 0) was nearly doubled in WWP2[Mut/Mut] (~35%) compared to WT mice (~16%), while the proportions of 'stem-cell like' (cluster 3) and 'Ly6c[high] monocytes' (cluster 4) were significantly reduced in WWP2[Mut/Mut] mice (Fig. 2b). Interestingly, the Ly6c[high] monocyte (cluster 4) infiltration is the main source of cardiac macrophage pool expansion (Fig. 1g and h), and cluster 4 is the most significantly affected cluster by WWP2[Mut/Mut] at day 7 following Ang-II-infusion (>80% reduction, Supplementary Fig. 4b).

We then investigated a potential link between WWP2 and the regulation of Ly6c[high] monocyte infiltrates in the heart. The expression of *Ly6c* was enriched in macrophage cluster 4 (Fig. 2c), and this surface glycoprotein was further used as a marker for flow cytometry analysis (Supplementary Fig. 4c). Ang-II-infusion significantly increased the percentage of Ly6c[high] cells in the cardiac monocyte compartment (CD45[+]CD64[+]CD11b[+33]) (Fig. 2d). Compared with WT, WWP2[Mut/Mut] showed reduced percentage of Ly6c[high] monocytes, which is in line with the scRNA-seq results (Fig. 2b). Consistent with the reduced Ly6c[high] cardiac monocyte pool, WWP2[Mut/Mut] showed less total resident cardiac macrophage numbers (CD45[+]CD64[+]CD11b[+]F4/80[+]Ly6G[−]; Fig. 2e, Supplementary Fig. 4g) and reduced percentage of cardiac macrophages within CD45+ cells at 7 days of Ang-II-infusion (Supplementary Fig. 4h). Systemically, the number of circulating neutrophils and monocytes were not affected in WWP2[Mut/Mut] mice (Supplementary Fig. 4d–f). Similarly, cell proliferation was not regulated by WWP2 (Supplementary Fig. 5a), and the WWP2 effect on the Ly6c[high] monocyte frequency (cluster 4) was not observed in the 'proliferating' macrophages (cluster 6) (Fig. 2b). In keeping with this, WWP2[Mut/Mut] bone marrow-derived macrophages (BMDMs) showed similar cell proliferation compared with WT mice (Supplementary Fig. 5b, c). Although WWP2[Mut/Mut] mice showed less CD68+ cells in cardiac sections when compared with WT mice, the proliferative CD68[+]Ki67[+] cells

did not show a significant difference between WWP2[Mut/Mut] and WT hearts after Ang-II-infusion (Supplementary Fig. 5d). Collectively, these findings indicate that WWP2 dysfunction attenuates Ly6c[high] monocyte infiltration and cardiac macrophage expansion in early-stage heart fibrosis, without affecting cell proliferation per se.

The 'Ly6c[high] monocyte' cluster presents relatively high *Ccr2* expressing cells (Supplementary Fig. 3b), with the ability to infiltrate tissues through chemotaxis, a process that directs cell migration. We next investigated chemokine/receptor pairs (Ccl-Ccr) using CellphoneDB[54], which identified significant Ccl-Ccr interactions between cardiac macrophage clusters (Supplementary Fig. 6a). Specifically, the 'Ly6c[high] monocyte cluster' accounted for the main interacting Ccr partners (e.g., *Ccr1*, *Ccr2,* and *Ccr5*) in the heart (Supplementary Fig. 6b). When compared with steady-state phase (saline control), Ang-II treatment significantly upregulated several Ccl-Ccr interaction pairs in cardiac macrophages (Fig. 2f), including chemokines such as *Ccl12* (aka monocyte-chemotactic protein 5, MCP5), *Ccl5* (aka RANTES), and *Ccl7* (aka MCP3). These Ccl-Ccr interactions between WT cardiac macrophages were almost all significantly downregulated in cardiac macrophages from WWP2[Mut/Mut] mice. The exception to this finding was *Ccl24-Ccr2*, the only Ccl-Ccr pair upregulated in WWP2[Mut/Mut] macrophages and primarily expressed in the 'homeostasis/repair' macrophages (cluster 0) (Fig. 2f, g).

*Ccl5* was found to be specifically expressed in 'Ly6c[high] monocytes' and almost absent in WWP2[Mut/Mut] cells, while other chemokines (*Ccl7* and *Ccl12*) showed wider expression throughout the clusters which was also WWP2-dependent (Fig. 2g). Ccl5 has been previously reported to play an important role in mononuclear cell recruitment and activation[55,56]. To validate the regulation of Ccl5 by WWP2, we sorted cardiac macrophages and measured their chemokine expression by qRT-PCR. Strikingly, Ang-II-infusion greatly upregulated *Ccl5* mRNA levels in WT cardiac macrophages, showing ~50-fold upregulation when compared to *Ccl7*, *Ccl12,* and *Ccl24* (Fig. 2h). WWP2[Mut/Mut] had significantly decreased mRNA levels of *Ccl5* compared to Ang-II induced WT cardiac macrophages (Fig. 2h). We further cultured BMDMs and spleen-derived macrophages (SPMs), as the bone marrow and spleen are the primary sites of production of circulating monocytes, which migrate, settle and differentiate into macrophages within inflamed sites[57–59]. As expected, pro-inflammatory stimuli (LPS/IFNγ) increased the expression levels of *Ccl5* in WT BMDMs and SPMs. When compared with WT, the levels of CCL5 were significantly reduced in WWP2[Mut/Mut] macrophages at both mRNA and protein levels, independently of their origin (bone marrow or spleen) (Fig. 2i, j). The secreted levels of Ccl5 in vitro were also significantly downregulated in WWP2[Mut/Mut] BMDMs, with respect to WT macrophages following LPS/IFNγ stimulation (Fig. 2k).

Taken together, our results show that WWP2 regulates the Ccl5/Ly6c[high] monocyte axis in fibrotic hearts at early phase (7 days post Ang II-infusion). WWP2 dysfunction suppresses *Ccl5* expression and reduces the cardiac Ly6c[high] infiltration, while sustaining the 'homeostasis/repair' macrophages. This is associated with a reduction of macrophage infiltrates in the heart.

## WWP2 promotes the inflammatory and profibrotic activity of cardiac macrophages

We next investigated how WWP2 regulates macrophage polarization beyond the Ccl5/Ly6c[high] monocyte infiltration axis. Compared with WT control, Ang-II-infusion changed the expression of 219 genes (Fig. 3a, *x* axis). Gene set enrichment analysis (GSEA) showed that these differentially expressed genes (DEGs) in cardiac macrophages were significantly enriched for three main pathways: pro-inflammatory response, ECM backbone, and PI3K/AKT signaling (Supplementary Fig. 7). The presence of the ECM backbone pathway confirmed that day 7 after Ang-II-infusion is an early fibrogenic time point during the time-course of NICM[46]. Strikingly, WWP2[Mut/Mut] broadly reversed Ang-II-

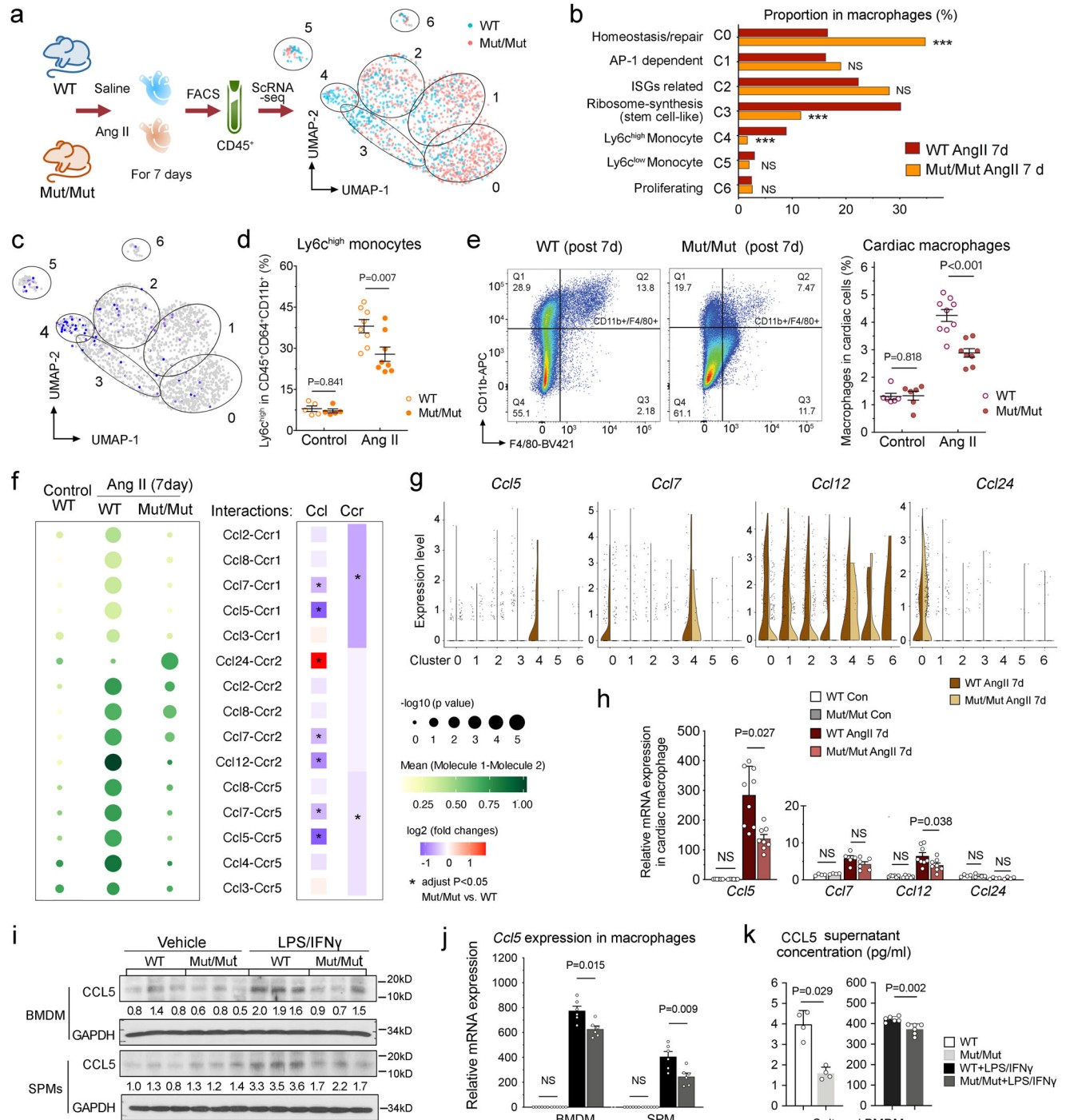

**Fig. 2 | WWP2 dysfunction regulates macrophage infiltration and expansion in the heart at 7 days post Ang II-infusion. a** Schematic representation of the experimental setup. Sorted live CD45+ cells from LV underwent scRNA-seq analysis, and seven clusters are represented in the UMAP space. WT, WWP2[wt/wt]; Mut/Mut, WWP2[Mut/Mut]. In each experiment, mice were treated with either saline (control) or Ang-II (500 ng/kg/min, 7 days). **b** Bar plots represent the proportions of macrophage clusters (Fig. 2a) in WT and Mut/Mut following Ang-II treatment. Two-proportion $z$ test, ***$P < 0.001$ Mut/Mut *vs* WT. **c** Expression of *Ly6c* gene in cardiac macrophages represented in the UMAP space. **d** Percentage of Ly6c[high] cells in WT and Mut/Mut cardiac monocyte compartments, in control or Ang-II-treated mice. $n = 5–8$ for each group. **e** Representative flow cytometry for CD11b[+]/F4/80[+] gate in Ang-II-treated WT and Mut/Mut mice (left). Quantification of cardiac macrophages in Mut/Mut compared with WT mice following Ang-II treatment (right). $n = 6–9$ for each group. **f** Chemokine ligand-receptor pairs (Molecule 1–2 pairs) derived by CellPhoneDB analysis in cardiac macrophages, $P < 0.01$ (left). Mean expression of

Molecule 1–2 pairs are reported as color intensity, and statistical significance ($-\log_{10}(p$ value)) as bubble size. Differential expression of chemokine ligands and receptors is represented as $\log_2$fold changes (FC) (*right*). Two-sided Wilcoxon test. *, Benjamini–Hochberg adjusted $P < 0.05$. **g–h** Chemokine ligand (*Ccl5, Ccl7, Ccl12, Ccl24*) mRNA expression in macrophage clusters from scRNA-seq analysis, and in cardiac macrophages sorted from LVs of control or Ang-II-treated WT and Mut/Mut mice ($n = 4–9$ for each group). **i–j** Representative western blot (**i**) and quantitative RT-PCR (qRT-PCR) (**j**) measuring Ccl5 in bone marrow-derived macrophages (BMDMs) and spleen-derived macrophages (SPMs) under LPS (100 ng/ml) and IFNγ (10 ng/ml) treatment (4 hrs) in WT and Mut/Mut experimental groups. Non-parametric Mann–Whitney U test, $n = 6$ for each group. **k** Levels of Ccl5 in the supernatant from WT and Mut/Mut BMDMs treated with LPS (100 ng/ml) and IFNγ (10 ng/ml) (4 hrs). $n = 5–6$ for each group. Unless otherwise indicated, data are shown as dot-plots with mean ± SD, and statistical significance is assessed by the non-parametric Mann–Whitney U test.

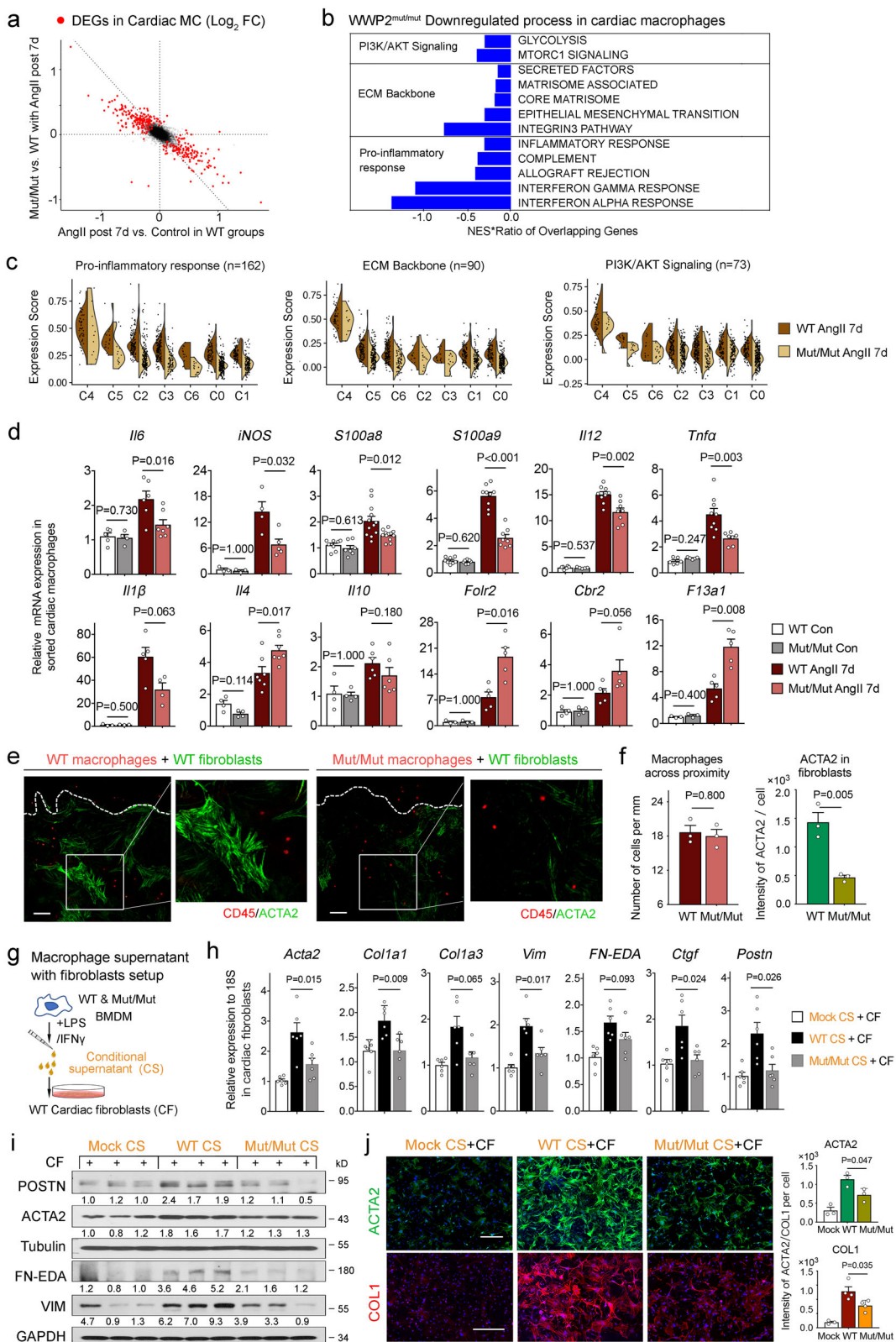

induced transcriptomic changes at the early phase of fibrogenesis (Fig. 3a, *y* axis). The expression of the three Ang-II-induced pathways (i.e., pro-inflammatory response, ECM backbone, and PI3K/AKT signaling) was significantly downregulated in WWP2^Mut/Mut cardiac macrophages post 7-day Ang-II-infusion (Fig. 3b). The most strongly suppressed pathway by WWP2 is the pro-inflammatory response, including interferon signature pathways (Fig. 3b).

We deconvoluted the NICM-induced pathways into the macrophage subpopulations and found that pro-inflammatory response was enriched in Ly6c^high monocytes and ISG clusters (clusters 4 and 2), while the ECM backbone and PI3K/AKT signaling were only enriched in Ly6c^high monocytes (Supplementary Fig. 8). Compared with WT, WWP2^Mut/Mut cardiac macrophages exhibited lower expression score of these three pathways, which are highly expressed in Ly6c^high

**Fig. 3 | WWP2 regulates macrophage activation and profibrotic function.**
**a** Scatter plot of $\log_2$fold changes (FC) in mRNA expression from scRNA-seq in cardiac macrophages between Ang-II-treated Mut/Mut and WT mice ($y$ axis) and $\log_2$FC between WT Ang-II-treated and WT untreated mice ($x$ axis). Differentially expressed genes (DEGs) in red ($n = 237$, FDR < 0.05). Ang-II treatment: 500 ng/kg/min, 7 days. **b** Top downregulated pathways in Mut/Mut macrophages identified by gene set enrichment analysis (GSEA) of DEGs. NES, normalized enrichment score. **c** Violin plots illustrate the expression score of the GSEA-derived pathways across all cardiac macrophage clusters in Mut/Mut and WT mice after treatment with Ang-II (7 days). **d** qRT-PCR analysis of selected pro-inflammatory and homeostatic/reparatory genes in macrophages sorted from LV of WT and Mut/Mut mice treated with saline or Ang-II (7 days). $n = 5$–12 for each group. **e** Representative immunofluorescence staining of smooth muscle aortic alpha-actin (ACTA2, green) in (myo)fibroblasts co-cultured with CD45 + macrophages (red). Scale bar, 100 μm. **f** Number of cardiac macrophages moving across the proximity border (left), and ACTA2 expression in (myo)fibroblasts (right). $n = 3$ per experimental group and 15–25 fibroblast images were taken from each slide. **g** Schematic of the co-culture experimental setup in vitro. The conditioned supernatant (CS) from BMDMs treated with LPS (100 ng/ml, 4 hrs) and IFNγ (10 ng/ml, 4 hrs) was used to activate primary cardiac fibroblasts (P2) cultured from LV of WT mice for 72 hrs. **h, i** Relative mRNA expression (**h**) and representative WB (**i**) of selected extracellular matrix (ECM) genes in cardiac (myo) fibroblast treated with CS from mock, WT and Mut/Mut BMDMs. $n = 6$ for each group. Mock, CS from untreated BMDMs. **j** Representative microscopy images (left) with immunostaining for ACTA2 and COL1 in cardiac fibroblasts after stimulation with CS from mock, WT, or Mut/Mut BMDMs. Scale bars, 500 μm. Bar plot (right) showing fluorescence intensity of ACTA2 and COL1 per fibroblast in the different experimental groups (mock, WT, Mut/Mut). $n = 3$–4 per experimental group, and 17–53 fibroblast images were taken from each slide. Unless otherwise indicated, data are shown as dot-plots with mean ± SD, and statistical significance is assessed by the non-parametric Mann–Whitney $U$ test.

monocytes (C4, Fig. 3c). In addition, WWP2$^{Mut/Mut}$ moderately upregulated homeostasis, and calcium and acid metabolism processes in cardiac macrophages post 7-day Ang-II-infusion (Supplementary Fig. 9a). These pathways are significantly enriched in 'homeostasis/repair' cluster (C0) (Supplementary Fig. 9b–i).

The dual modulatory action of WWP2 on cardiac monocytes/macrophages (i.e., reduction of pro-inflammatory and induction of homeostasis/repair states) suggests a potential effect on macrophage polarization based on M1/M2 classification. Although simplistic and context-dependent[60–62], the M1-like (pro-inflammatory or classical) or M2-like (alternative or anti-inflammatory and repair) cardiac macrophages have been broadly described as Ccr2+ and Ccr2− cells, respectively[63]. In accordance with prior data categorizing cardiac macrophage subsets according to *Ccr2* expression (Supplementary Fig. 3d, e), we observed an 'M1-like' gene signature throughout all the subclusters but more prominently in Ccr2$^{high}$ cluster 4 (Ly6c$^{high}$ monocytes), while the 'M2-like' signature was present in all subclusters but in relatively high proportion within the homeostasis/repair cluster (Supplementary Fig. 10, and details in Methods). Notably, WWP2 dysfunction downregulated the M1-like signature, while upregulating M2-like signature (Supplementary Fig. 10). While this shift from M1-like to M2-like affected all subclusters, it was more prominent in the pro-inflammatory (clusters 4 and 2 associated with reduction of M1-like signature) and homeostasis (cluster 0 associated with increased M2-like) clusters.

To validate the scRNA-seq results, we measured the expression of representative M1/M2-like genes in cardiac macrophages sorted from hearts following Ang-II-infusion (Fig. 3d, CD45$^+$CD64$^+$CD11b$^+$F4/80$^+$Ly6G$^−$). Compared with WT macrophages, WWP2$^{Mut/Mut}$ had reduced mRNA levels of *S100a9, Il12, Il6, Nos2 (iNOS)*, and increased expression of *Il4* and *F13a1*. We also analyzed in vitro-differentiated WT and WWP2$^{Mut/Mut}$ macrophages stimulated with LPS/IFN-γ (i.e., classical activation). Compared with WT BMDMs, the reduced M1-like signature was confirmed both at gene and protein levels in WWP2$^{Mut/Mut}$ cells, which also showed decreased levels of secreted IL6 (Supplementary Fig. 11). A similar anti-inflammatory effect of WWP2$^{Mut/Mut}$ was observed in classically-activated SPMs (Supplementary Fig. 12). Furthermore, IL-4/IL13 or TGFβ1 treatment (i.e., alternative activation) increased the mRNA levels of homeostatic genes in both BMDMs and SPMs (Supplementary Fig. 13). WWP2 dysfunction further increased the expression of M2-like genes, such as CD206 (*Mrc1*), *Il10, Il4, Folr2*, and *Cbr2*, in BMDMs upon TGFβ1 treatment (Supplementary Fig. 13).

We report robust expression of transcripts belonging to ECM backbone pathway that is under WWP2 control (Fig. 3b, c). Interestingly, the ECM pathway was identified specifically to the 'Ly6c$^{high}$ monocyte' (Supplementary Fig. 8), which is controlled by WWP2 at two levels: infiltration through Ccl5 (Fig. 2g) and pro-inflammatory activation (Fig. 3c). To explore the fibrogenic properties of WWP2 in more detail, we further tested the potential contribution of WWP2 to the macrophage-fibroblast crosstalk, using an ex vivo co-culture system (Supplementary Fig. 14a). Following 72 hours of co-culture, ex vivo isolated cardiac macrophages from NICM mice migrated toward primary cardiac fibroblasts (Supplementary Fig. 14b). The expression of smooth muscle Actin Alpha 2 (ACTA2) was increased in primary fibroblasts, especially within the boundary area where the cells are in contact with macrophages. Using such ex vivo co-culture system, we observed similar numbers of WT and WWP2$^{Mut/Mut}$ cardiac macrophages (*in red*) in the vicinity of fibroblasts. However, ACTA2 protein expression was significantly reduced in fibroblasts co-cultured with WWP2$^{Mut/Mut}$ macrophages compared with WT macrophages (Fig. 3e, f). In addition to cell-to-cell interaction, macrophages regulate fibroblast activation through secretion of soluble factors. We tested this possibility by performing supernatant transfer experiments using BMDMs (Fig. 3g). Conditioned supernatant (CS) from WT BMDMs stimulated with LPS/IFNγ increased ECM production at mRNA and protein levels in primary cardiac fibroblasts, (Fig. 3h–j). Importantly, the CS from WWP2$^{Mut/Mut}$ BMDMs resulted in significant rescue of myofibroblast differentiation (Fig. 3h–j). Using the mean fluorescence intensity (MFI) per fibroblast, immunofluorescence staining of ACTA2 and COL1 confirmed a relatively less pronounced fibrotic response in fibroblasts treated with WWP2$^{Mut/Mut}$ CS (Fig. 3j).

In summary, WWP2 dysfunction robustly suppresses pro-inflammatory activation of cardiac macrophages at early phase of cardiac fibrosis. This inhibitory effect is mainly on the Ly6c$^{high}$ monocyte subpopulation where WWP2 regulates pro-inflammatory, M1-like gene signature. WWP2 dysfunction also limits the profibrotic potential of cardiac macrophages ex vivo, suggesting the regulatory function of WWP2 in the transition from inflammatory to fibrotic phase during the course of NICM.

## WWP2 dysfunction in monocytes/macrophages protects against cardiac fibrosis

Besides the regulatory function of WWP2 in cardiac macrophages[39], the contribution of myeloid WWP2 to cardiac fibrosis remained to be established in vivo. To address the effect of WWP2 in monocyte/macrophage lineage in vivo, we generated a murine model of WWP2$^{flox/flox}$LysM$^{cre}$ (WWP2$^{Mac}$) using the Cre-lox system (Supplementary Fig. 15a). The deletion of the exon 3 flanked by loxP sites was generated by crossing WWP2$^{flox/flox}$ mice with a transgenic line expressing the recombinase (LysM$^{cre}$). Cre-mediated removal of floxed *WWP2* was confirmed by western blot (WB) and qPCR analyses in cardiac, spleen, and bone marrow-derived macrophages (Supplementary Fig. 15b–d) from WWP2 conditional knockout mice (WWP2$^{Mac}$). The expression of WWP2 was also reduced in heart tissue but not changed in cultured cardiac fibroblasts (Supplementary Fig. 15e, f). Unlike the global *Wwp2* mutant mice (WWP2$^{Mut/Mut}$)

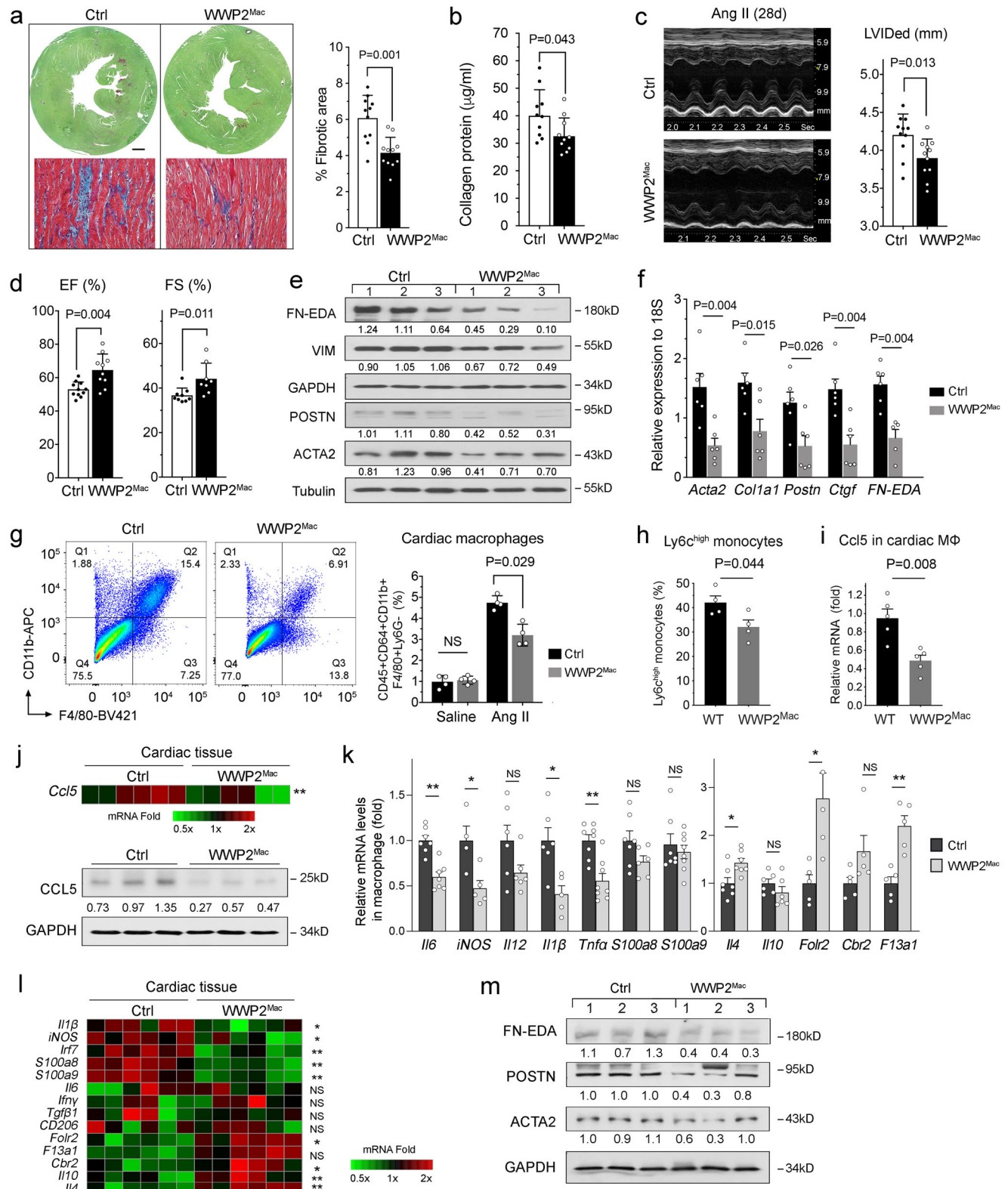

presenting relatively reduced body weight, abnormal craniofacial development, and elongated teeth[39], WWP2[Mac] mice showed only a non-significantly reduced body weight ($P = 0.270$) (Supplementary Fig. 15g).

Using the WWP2[Mac], we next employed the same Ang-II-infusion model to test whether deletion of WWP2 in the myeloid lineage confers protection against cardiac fibrosis in NICM, which was observed in the global WWP2 dysfunction model[39]. As expected, post 28-day Ang-II-

infusion resulted in increased tissue fibrosis, ventricular remodeling, and worsened cardiac function in WWP2[flox/flox] (Ctrl) mice. WWP2[Mac] mice showed a significant reduction of cardiac fibrosis, as shown by the lower percentage of fibrotic area by histopathology, accompanied by an attenuation of collagen accumulation (Fig. 4a, b). Both cardiac hypertrophy and function were improved in WWP2[Mac] mice, which showed decreased left ventricular internal diameter (LVIDed) and increased fractional shortening (FS%) and ejection fraction (EF%) when

**Fig. 4 | Conditional depletion of WWP2 in macrophages protects from Angiotensin II induced cardiac fibrosis. a** Left, representative Sirius red (upper) and Masson's Trichrome (lower) staining of short-axis sections of LV. Scale bar: 0.5 mm. Right, quantification of the area of fibrosis in transverse histological sections with Sirius red staining at the mid-ventricular level. WWP2$^{Mac}$, WWP2 conditional knockout in macrophages; Control (Ctrl), WWP2$^{flox/flox}$ mice. **b** Collagen content in LV by hydroxy-phenyl-propionic acid (HPA) assay. $n = 10$ per experimental group. **c** Representative M-mode echocardiograms (middle LV long-axis) (left), and quantification of left ventricular internal dimension at end-diastole (LVIDed)(*right*). $n = 12$ per experimental group. **d** Echocardiogram-based quantification of LV ejection fraction (EF%) and fractional shortening (FS%). $n = 8–11$ per experimental group. **e** Representative WB showing fibronectin extracellular domain A (EDA-FN), vimentin (VIM), periostin (POSTN), and ACTA2 in LV tissue from control and WWP2$^{Mac}$ mice. **f** qRT-PCR analysis of selected ECM genes in LV tissue from control and WWP2$^{Mac}$ mice. $n = 6$ per experimental group. **g** Representative flow cytometry for CD11b$^+$/F4/80$^+$ gate for live cells from LV of control and WWP2$^{Mac}$ mice (left).

Quantification analysis shows reduced percentage of cardiac macrophage (live, CD45$^+$CD64$^+$CD11b$^+$F4/80$^+$Ly6G$^-$) in WWP2$^{Mac}$ compared with control mice (right). $n = 4$ for each group. **h** Percentage of the Ly6C$^{high}$ in cardiac monocyte compartments (CD45$^+$CD64$^+$CD11b$^+$). $n = 4$ for each group. **i** Ccl5 mRNA expression in cardiac macrophages (MΦ) sorted from LV. $n = 5$ per experimental group. **j** Relative mRNA levels of Ccl5 (upper), and representative WB for protein levels of CCL5 (lower) in LV tissue from control and WWP2$^{Mac}$ mice. **k–l** qRT-PCR analysis of selected pro-inflammatory and homeostatic/reparatory genes in sorted cardiac macrophages (**k**: bar plot, $n = 4–8$ per group) and in LV tissue (**l**: heatmap, $n = 6$ per group). **m** ECM protein expression in LV from control and WWP2$^{Mac}$ mice. Data in **a**–**f** refer to mice treated with Ang-II (500 ng/kg/min, 28 days). Data in **g**–**m** refer to mice treated with Ang-II (500 ng/kg/min, 7 days). Unless otherwise indicated, data are shown as dot-plots with mean ± SD, and statistical significance is assessed by the non-parametric Mann-Whitney $U$ test. WWP2$^{Mac}$ *vs* Ctrl: *$P < 0.05$; **$P < 0.01$; NS, not significant.

---

compared with Ctrl mice (Fig. 4c, d). The myocyte mean cell area was reduced in WWP2$^{Mac}$ hearts compared to WT controls following 7 days Ang-II-infusion, while left ventricular mass index (LVMI) showed only a trend towards reduction (Supplementary Fig. 16a, b). Protein and mRNA analyses of representative ECM markers, such as ACTA2, COL1A1, and POSTN, showed that WWP2$^{Mac}$ hearts had significantly reduced fibroblast activation (Fig. 4e, f).

We further evaluated the effect of monocyte/macrophage-specific deletion of WWP2 on cardiac inflammation during the early fibrotic phase. Consistent with global WWP2 dysfunction (Fig. 2e), WWP2$^{Mac}$ reduced significantly cardiac macrophage expansion at day 7 following Ang-II-infusion (Fig. 4g). Within the cardiac monocyte compartment (CD45$^+$CD64$^+$CD11b$^{+33}$), WWP2$^{Mac}$ significantly reduced the percentage of Ly6c$^{high}$ monocytes (Fig. 4h), but had no effect on circulating neutrophils and monocytes and cardiac immune cells, such as neutrophils, T cells and B cells (Supplementary Fig. 16c–e).

Furthermore, sorted macrophages from WWP2$^{Mac}$ mice showed lower mRNA levels of Ccl5 than WT control cells (Fig. 4i). Ccl5 mRNA and protein levels were also reduced in the heart tissue of WWP2$^{Mac}$ mice (Fig. 4j). These findings suggested that monocyte recruitment to the heart was perturbed in the absence of WWP2 in myeloid cells. Unlike the normotensive controls (Supplementary Fig. 16f), Ang-II infused WWP2$^{Mac}$ mice had significantly reduced levels of M1-like genes, and increased M2-like genes in macrophages isolated from hearts when compared with control cells (Fig. 4k). These effects of WWP2$^{Mac}$ were conserved in total cardiac tissue (Fig. 4l). In addition, WWP2$^{Mac}$ hearts showed reduced POSTN and EDA-FN protein levels, confirming that WWP2 drives early pro-fibrogenic responses (Fig. 4m).

In summary, WWP2 deletion in monocytes/macrophages is sufficient to produce antifibrotic and cardiac protective effects. These effects are partly mediated through WWP2's dual action on macrophage infiltration and activation at the early fibrotic phase of the injury, including Ccl5-dependent recruitment of monocytes, and regulation of M1-like pro-inflammation process in cardiac macrophages.

## WWP2 modulates IRF7 signaling in cardiac macrophages

WWP2 regulatory effects on monocyte/macrophage infiltration/polarization pointed toward Ccl5, a chemokine that initiates inflammatory responses through monocyte infiltration to the heart. The mechanisms through which WWP2 regulates Ccl5 expression and, to a larger extent, pro-inflammatory and fibrotic responses remain to be elucidated. To this end, we sought to identify potential targets downstream of WWP2 during early heart fibrosis. pyScenic analysis of scRNA-seq data showed that WWP2 significantly regulated transcription factor networks (regulons) and in particular, an interferon regulatory factor 7 (IRF7)-driven regulon ($n = 987$ genes) which was the most significantly downregulated regulon in WWP2$^{Mut/Mut}$ (adjusted $P < 10^{-50}$) (Fig. 5a). The IRF7 regulon is mostly enriched within the

proinflammatory Ly6c$^{high}$ monocytes (C4) and ISG cluster (C2), the clusters previously established to drive Ccl5-dependent monocyte recruitment, M1-like activation, and ECM backbone pathways (Figs. 2g, 3b, c, Supplementary Figs. 7, 9). All these pathways are primarily regulated by WWP2, and WWP2 dysfunction results in reduced IRF7 regulon activity in the majority of macrophage subclusters (Fig. 5b). Since BMDMs cultured from WWP2$^{Mac}$ lack WWP2 (Supplementary Fig. 14), these WWP2$^{-/-}$ cells (shown as −/−) were used in mechanistic studies aiming to link WWP2 to IRF7.

IRF7, together with other IRF family members, is a master regulator of interferon-dependent immune responses[64–66]. Of note, 34% (80/237) of the DEGs regulated by WWP2 in cardiac macrophage (Fig. 5a) are targets of IRF7, and are significantly enriched for innate immune processes (hypergeometric test, $P = 8.65 \times 10^{-39}$) (Fig. 5c). We confirmed this functional enrichment using another database (Gene Transcription Regulation Database, GTRD), which identified 125 out of the 237 (53%) WWP2-regulated DEGs as IRF7 targets (Supplementary Fig. 17a). We further validated that the mRNA and protein levels of type I and II IFN signaling genes (e.g., Ifna, Ifnb, and Ifng) are regulated by WWP2 in BMDMs (Fig. 5d, e).

To confirm the enrichment for IRF7 regulon genes among WWP2 targets, we further tested whether these genes have an IRF7 binding site (Fig. 5f). Using previously published IRF7 ChIP-seq data generated in BMDMs[67], we identified 3 and 4 IRF7 binding site motifs in Ccl5 and Irf7, respectively (Fig. 5g). These were located around the genes' transcription start site (TSS) (Supplementary Fig. 17b). Our previous data confirmed that macrophage Ccl5 is a WWP2 target (Fig. 2g–k, Fig. 4i, j). Moreover, here we show the same regulatory effect of WWP2 with regard to Irf7 at both mRNA and protein levels in BMDMs (Fig. 5h, i). Importantly, IRF7 ChIP-qPCR in BMDMs showed that the absence of WWP2 significantly reduces the binding of IRF7 to Ccl5 and Irf7 loci in LPS/IFNγ stimulated cells (Fig. 5j). These findings suggest that WWP2 regulates cardiac macrophage function through Irf7 signaling upstream of the Ccl5-mediated recruitment of Ly6c$^{high}$ monocytes.

In order to characterize the WWP2-mediated control of IRF7, we tested a possible interaction between WWP2 and IRF7. WWP2 protein is upregulated by LPS treatment and is found in the cytoplasm of BMDMs, suggesting its cytoplasmic regulatory function in macrophages (Supplementary Fig. 18a, b). WWP2 immunoprecipitation following LPS stimulation in BMDMs showed direct binding of WWP2 to IRF7, but not to phosphorylated-IRF7 (p-IRF7) (Fig. 6a, *upper*). The IRF7-WWP2 interaction was confirmed by IRF7 immunoprecipitation followed by WWP2 blotting in activated BMDMs (Fig. 6a, *lower*). Using exogenously expressed FLAG-WWP2 fusion proteins, we confirmed an IRF7-WWP2 interaction that mainly involves the WWP2-N isoform (Supplementary Fig. 18c). Confocal microscopy analysis further showed colocalization of IRF7 and WWP2 in the cytoplasm of BMDMs (Mander's Overlap Coefficients (MOC) of 0.96 ± 0.02 and

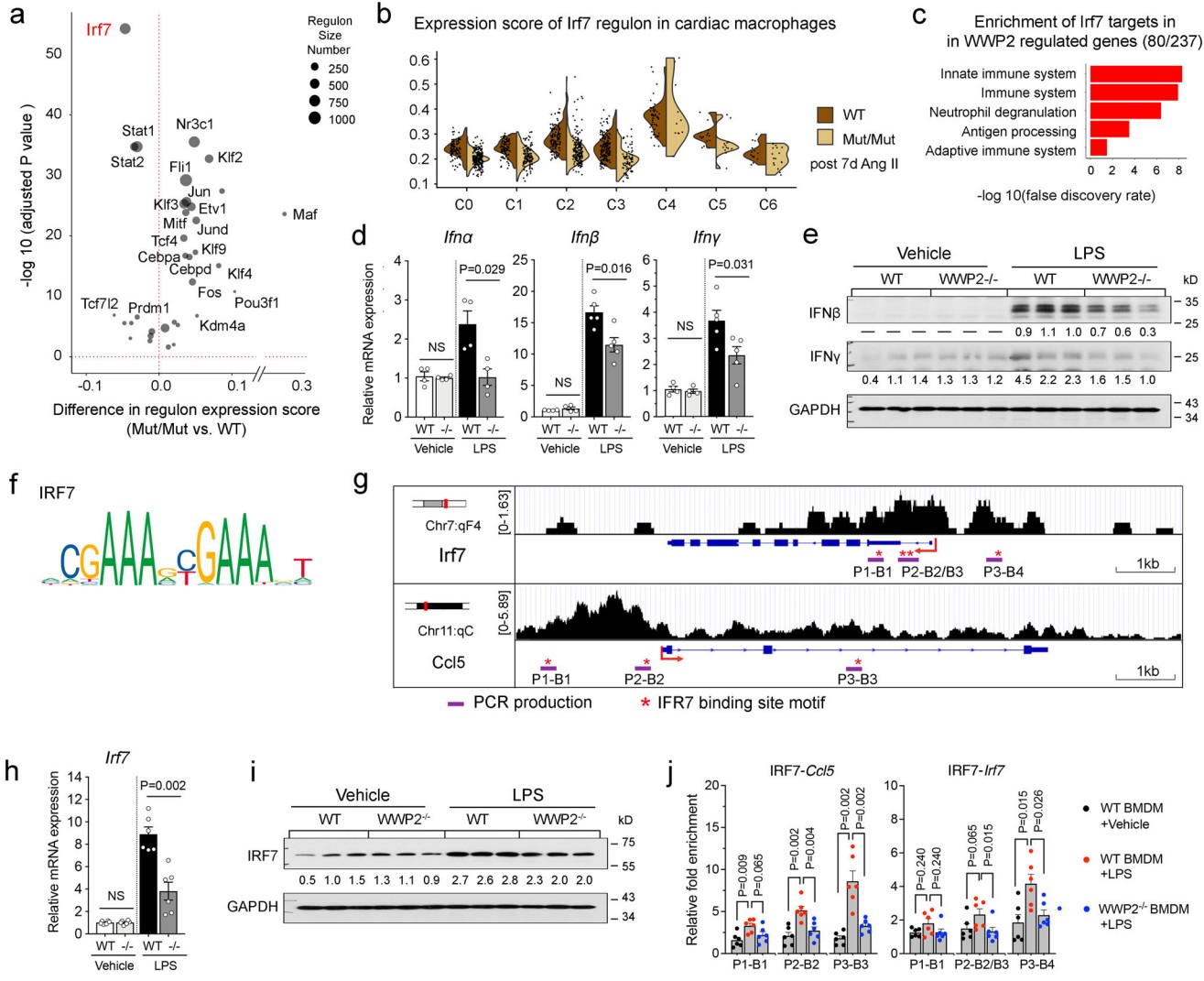

**Fig. 5 | WWP2 regulates IRF7 network activity in macrophages in NICM.**
**a** Scatter plot of the relative activity of transcription factor (TF)-regulons in cardiac macrophages from WWP2[Mut/Mut] (Mut/Mut) and WWP2[wt/wt] (WT) mice treated with Ang-II (7 days). For each regulon, the difference in expression score between Mut/Mut and WT and its statistical significance score are shown on the *x* axis and *y* axis, respectively. The number of downstream target genes in each regulon is proportional to the circle size (more details in Methods). The IRF7 regulon showed the largest and most significant expression score difference between Mut/Mut and WT cardiac macrophages. **b** Average expression score of the IRF7 regulon across cardiac macrophage clusters from WT and Mut/Mut mice treated with Ang-II (7 days). (*P* values by two-sided Wilcoxon test). **c** Breakdown of IRF7 targets enrichment in the profibrotic hECM-network (*n* = 237 genes) previously identified in LV of dilated cardiomyopathy patients[39]. 80 (33.8%) of these hECM-network genes are targets of IRF7. **d**, **e** qRT-PCR (**d**) and WB (**e**) of representative IFN signaling genes (IFNα, IFNβ,

and IFNγ) in WT and WWP2[−/−] (−/−) BMDMs with and without LPS stimulation. *n* = 5 per experimental group. **f**. IRF7 binding site motif derived from previously published IRF7 ChIP-seq analysis[67]. **g** IRF7 ChIP-qPCR analysis and schematic showing the IRF7 binding site motifs (*P* < 10[−4]) within the *Ccl5* and *Irf7* genes. *Motifs matching to (*Ccl5*, *Irf7*) PCR products (purple thick line), which were used in the ChIP-qPCR analysis. **h**,**i** qRT-PCR (**h**) and WB (**i**) of IRF7 mRNA and protein expression in WT and WWP2[−/−] (−/−) BMDMs. *n* = 3–6 for each group. **j** ChIP-qPCR analysis of IRF7 occupancy sites on *Ccl5* (left) or *Irf7* (right) in WT and WWP2[−/−] BMDMs treated with LPS. Enrichment is normalized to input DNA, and represented as fold-enrichment relative to vehicle controls in WT BMDMs. In each case, LPS stimulation (100 ng/ml, 4 hrs). Unless otherwise indicated, data are shown as dot-plots with mean ± SD, and statistical significance is assessed by the non-parametric Mann–Whitney *U* test.

Pearson's correlation coefficient of 0.89 ± 0.03 in the intensity profiles[68]) (Fig. 6b).

In addition to their physical interaction, we observed that WWP2 promotes the ubiquitination of IRF7. Ubiquitination assay followed by IRF7 immunoprecipitation detected several slower migrating bands in WT BMDMs treated with LPS, indicating the formation of mono-ubiquitin and multiple monoubiquitin or poly-ubiquitin chains (Fig. 6c). The level of ubiquitin was drastically reduced in WWP2[−/−] BMDMs, especially IRF7-Ub1, when compared with the highly ubiquitinated IRF7 in WT BMDMs. In addition to having relatively higher ubiquitination, WT BMDMs showed higher levels of IRF7 (Fig. 5h, i) and p-IRF7 (Fig. 6d).

We next measured the transcriptional activity of *Irf7*, which is a widely used proxy for IRF7 signaling activation. LPS stimulation recruited interferon regulatory factors (IRFs) to the *Irf7* interferon-sensitive response element (ISRE) in BMDMs transfected with vectors encoding ISRE luciferase, and significantly induced their transcriptional activation (Fig. 6e). LPS-dependent IRF7 reporter activity was found to be significantly lower in WWP2[−/−] BMDMs compared to WT cells (Fig. 6e). Transient knockdown of WWP2 expression by siRNA equally resulted in decreased transcriptional activity of IRF7 in NIH-3T3 cells (Supplementary Fig. 19). These findings indicate that WWP2 triggers IRF7 activation by non-degradative ubiquitination in macrophages.

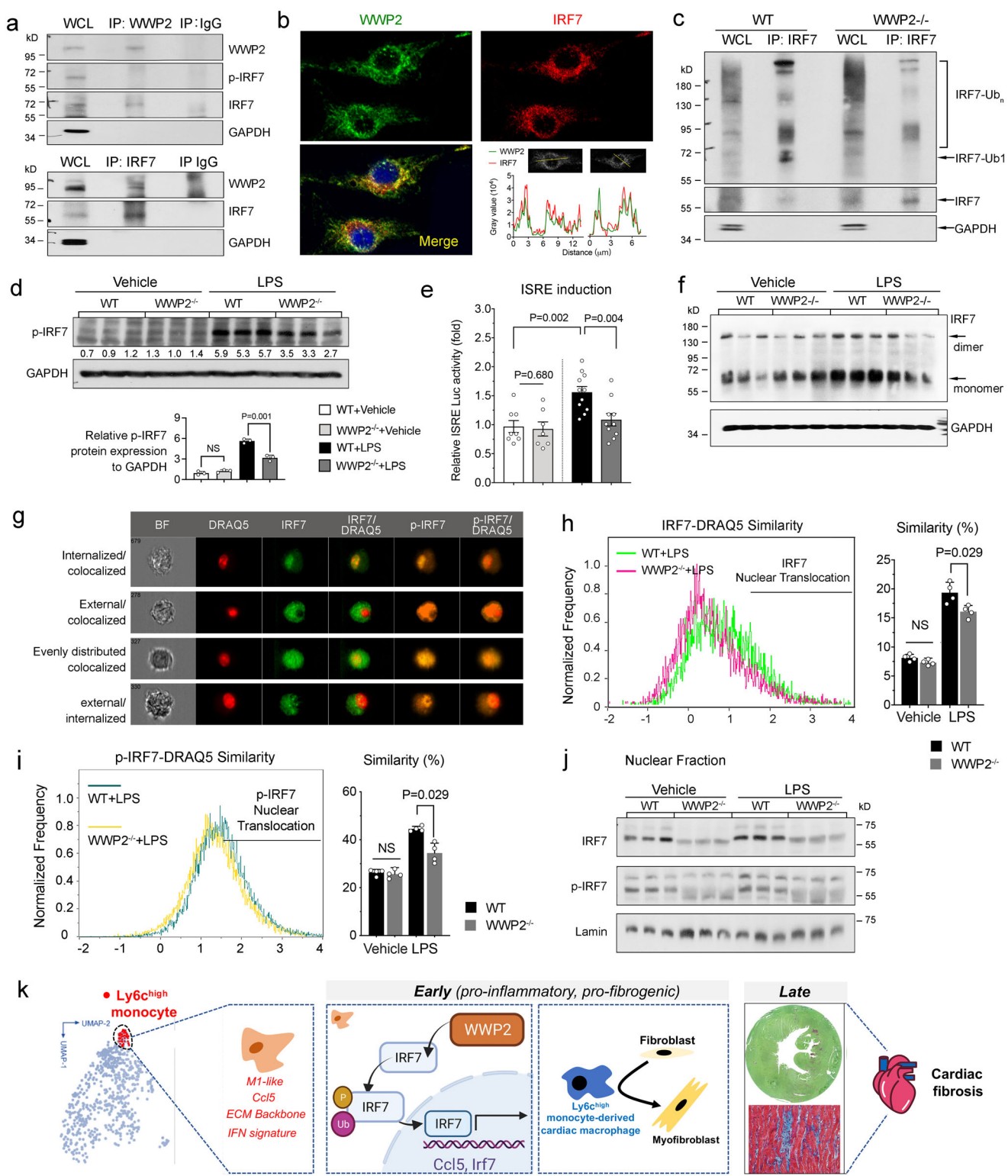

Transcriptional activation of IRF3/7 has been reported to be mediated by phosphorylation, dimerization formation, and subsequent translocation from the cytoplasm into the nucleus[69,70]. Compared with WWP2[−/−], the levels of p-IRF7 were significantly higher in WT BMDMs following LPS stimulation (Fig. 6d). Non-denaturing gels assay showed that LPS induced the formation of IRF7 dimers in WT BMDMs, but to a lesser extent in WWP2[−/−] cells (Fig. 6f). We show the colocalization of IRF7 and p-IRF7 inside (internalized) and outside (external) the nucleus in BMDMs (Fig. 6g), and the high similarity of both nuclear

IRF7 and p-IRF7 with DRAQ5 (Supplementary Fig. 20a, b). The percentage of nuclear IRF7 and p-IRF7 in WT BMDMs was increased by LPS stimulation, and this was significantly reduced in WWP2[−/−] BMDMs (Fig. 6h, i). Consistently, the expression of nuclear IRF7 and p-IRF7 were both decreased in WWP2[−/−] (Fig. 6j). The presence of fast-moving bands in IRF7 and p-IRF7 blots (Fig. 6j) suggests other forms of protein modification of IRF7 as previously described[71].

Taken together, our scRNA-seq analyses uncovered an IRF7-driven gene network regulated by WWP2 in cardiac macrophages

**Fig. 6 | WWP2 regulates IRF7 transcriptional activity in macrophages through non-degradative ubiquitination. a** Representative WB of co-immunoprecipitation experiment with WWP2 (upper) or IRF7 (lower) showing a direct interaction of WWP2 with IRF7 in BMDMs following LPS stimulation. WCL, whole cell lysate. **b** Expression of WWP2 (green) and IRF7 (red) in BMDMs following LPS stimulation. The intensity profiles of WWP2 and IRF7 were plotted along an ideal straight line (yellow) crossing the nuclei of two representative cells, showing the WWP2 and IRF7 signals overlap (bottom right). **c** In-cell ubiquitylation analysis of IRF7 in BMDMs from control WT and WWP2⁻/⁻ mice. Cells were treated with MG132 (10uM, 3 hrs) followed by LPS stimulation. **d** Representative WB measuring p-IRF7 protein expression in WT and WWP2⁻/⁻ BMDMs with or without LPS stimulation. Quantification of p-IRF7 protein expression is shown in the bar plot below. **e** Quantification of the IRF7 Interferon-stimulated response element (ISRE) luciferase activity in WT and WWP2⁻/⁻ BMDMs following LPS treatment. *n* = 7–11 for each group. **f** Representative immunoblots of native Polyacrylamide gel electrophoresis (PAGE) identifying monomeric and dimeric forms of IRF7 (indicated by arrows) in WT and WWP2⁻/⁻ BMDMs with or without LPS stimulation. **g** Representative cellular images (BMDMs) of in-cell IRF7 and p-IRF7 from 10,000 acquired events by imaging flow cytometry (see Methods), showing typical externalized and internalized patterns of colocalized or separately distributed IRF7 and p-IRF7. **h, i** Imaging flow cytometry shows IRF7 (**h**) and p-IRF7 (**i**) DRAQ5 similarity in WT and WWP2⁻/⁻ BMDMs following LPS treatment, using a similarity score cutoff ≥1.5 for protein nuclear translocation (left panels). Quantification of nuclear translocation of IRF7 and p-IRF7 (right panels). *n* = 4 for each group. **j** Representative WB showing IRF7 and p-IRF7 protein distribution in nuclear fractions of BMDMs (with or without LPS) from WT and WWP2⁻/⁻ mice. **k** Schematic of the proposed mechanism through which WWP2 regulates IRF7 and the Ccl5/Ly6c^high monocyte axis during the early phase of fibrogenesis, which in turn affects cardiac tissue fibrosis at a later stage of the fibrogenic process. In each experiment, LPS stimulation (100 ng/ml, 4 hrs). Unless otherwise indicated, data are shown as dot plots with mean ± SD, and statistical significance is assessed by the non-parametric Mann–Whitney *U* test.

7 days post Ang-II treatment. WWP2 modulates IRF7-mediated CCL5 and IFN signaling, both important for the infiltration and inflammatory response of cardiac monocytes/macrophages. WWP2 interacts with IRF7, promoting its non-degradative ubiquitination and further modulating its phosphorylation and dimerization. IRF7 nuclear translocation and transcriptional activity are subsequently reduced in activated macrophages lacking WWP2.

## Discussion

Non-ischemic heart disease arises from different aetiologies and hypertension is one of them. Hypertension can cause cardiac injury and fibrosis due to prolonged mechanical stress and impaired remodeling. Different models of NICM have previously reported increasing numbers of cardiac monocyte-derived macrophages in response to the experimentally-induced mechanical stress (pressure overload or hypertension)[11,17,72], but the functional characterization of these macrophages and their regulatory role in cardiac fibrosis has only been recently explored[38,52]. For instance, cardiac resident macrophage subsets respond in a different way to hypertensive stress[38], and bone marrow-derived Rel-mediated CD72^high macrophages are pro-inflammatory in mice with TAC[52]. These studies show how diverse macrophage subsets, whether resident or monocyte-derived, participate to cardiac remodeling and in the regulation of fibrosis during NICM.

In this study, we started from establishing the single-cell landscape of the Ang-II-infused mouse heart at day 7, a time point that corresponds to the peak of cardiac macrophages. We identify two cardiac macrophage subsets that are either homeostatic or monocyte-derived/pro-inflammatory with existence of different states of activation in between the two. Our findings are broadly in line with recent scRNA-seq analyses of cardiac macrophages in myocardial infarction[33], DCM[51], TAC[52], and Ang-II-infused hypertrophy[38]. They show that homeostatic/repair macrophages express high *Timd4*, and low *Ccr2* and *Cd72*. Conversely, the monocyte-derived/pro-inflammatory cluster expresses high *Ccr2*, *Cd72*, and low *Timd4*.

Absent in the basal condition (saline), the Ly6c^high monocyte cluster appears following Ang-II-infusion, and our cell trajectory analysis suggests it is a main contributor of the expansion of cardiac macrophages 7 days following hypertensive injury. We found that the ISGs-related macrophages are mainly derived from these Ly6c^high monocytes along the pseudotime trajectory, and both are enriched for *Ccr2*-expressing cells. Given that Ccr2 has been extensively validated as a cell surface marker of monocyte-derived cardiac macrophages[14,31,33,73], our results reveal the trajectory of Ly6c^high monocyte infiltration to the heart. On the other hand, Ang-II causes the reduction of 'homeostasis/repair' and 'AP-1' macrophages that express relatively low *Ccr2* but high levels of the anti-inflammatory 'M2-like' chemokine *Ccl24*[74].

Our results also highlight the functional properties of Ly6c^high monocytes during NICM. This subgroup of macrophages is enriched in unique chemokines (e.g., Ccl5), as well as pathways such as ECM backbone and PI3K/AKT; the latter being a regulator of monocyte chemotaxis, macrophage proliferation[75] and IRF7 nuclear translocation in TLR-stimulated plasmacytoid predendritic cells[76]. The enrichment of Ly6c^high monocytes for the ECM backbone pathway is of particular interest as it confirms previous results showing the pro-fibrogenic effects of Ang-II-infusion in the heart[45,46] by further delineating the macrophage subpopulation responsible for these effects. Thus, macrophage subpopulations seem to have antagonizing effects on cardiac fibrosis during NICM. Loss of cardiac resident macrophages correlates with increased fibrosis[38], whereas blocking infiltrating monocytes reduce fibrosis in Ang-II-induced cardiac hypertrophy[77]. In fact, Ly6c^high monocytes and F4/80+ macrophages are critical early mediators of Ang-II-induced cardiac fibrosis in mice[24,78,79]. More generally, Ly6c^high monocytes are precursors of monocyte-derived phagocytes in inflamed tissues[80–83], and the tissue microenvironment contributes to the monocyte-to-phagocyte transition that seems to be a key feature of Ly6c^high monocytes[84].

Our work identifies WWP2 as a critical regulator of cardiac monocyte/macrophage infiltration and activation in early-phase non-ischaemic fibrosis. We have previously shown that *WWP2* is a regulator of pathological cardiac fibrosis through the control of a profibrotic gene network conserved in rat and human heart disease characterized by diffuse myocardial remodeling and fibrosis[39]. The WWP2-regulated profibrotic gene network is conserved across different heart diseases[39], which suggests a potential wider regulatory role of *WWP2* including macrophage-mediated cardiac remodeling. Our previous work also showed that mutant WWP2 confers protection against cardiac fibrosis through the modulation of fibroblast function[39], but the relative contribution of immune cell WWP2 in this process required further clarification. Here we establish a major role of monocyte/macrophage WWP2 in regulating inflammation and fibrosis during NICM. While we cannot exclude a contribution of WWP2 through T-cell, B-cell, or neutrophil-mediated cell activation, WWP2 expressed by monocytes/macrophages is sufficient to mediate cardiac fibrosis through regulation of Ly6c^high monocyte infiltration and expansion.

Specifically, the regulatory effect of WWP2 is on CCL5 secretion which is mostly restricted to the Ly6c^high monocytes. Indeed, WWP2^Mut/Mut phenotype was associated with *Ccl5* lacking in the Ly6c^high monocyte compartment. Ex vivo isolated cardiac macrophages following Ang-II-infusion, showed a dramatic increase in *Ccl5* expression which was reduced in WWP2^Mut/Mut animals. Ccl5 is critical for macrophage recruitment and infiltration to tissues[56,85–87], and the CCL5-mediated macrophage recruitment associates with myocardial infarction[88] and cardiac dysfunction[89]. Our results do not suggest a role for WWP2 in monocyte cell survival, and this is consistent with lack of WWP2-regulated pathways related to cell death or apoptosis

in the cardiac macrophage scRNA-seq dataset. However, a WWP2 effect on chemoattractants other than Ccl5 cannot be ruled out.

At the pathway level, WWP2 dysfunction caused a reduction of Ly6c[chi] monocyte-specific pathways (ECM backbone and PIK3/AKT) with a robust down-regulation of interferon signature pathway, which was shared between the pro-inflammatory Ly6c[high] monocyte and ISG macrophage clusters. Loss of function of WWP2 was generally associated with a shift from M1-like to M2-like macrophage phenotype in vitro, suggesting that WWP2 is a regulator of macrophage polarization. The single-cell deconvolution of this functional shift in macrophages showed that WWP2 regulates pro-inflammatory clusters more profoundly compared to homeostasis/repair clusters. Recently, single-cell transcriptomics strategies allowed in situ measurement of transcription factor-mediated differentiation and heterogeneity of monocytes/macrophages[90]. In keeping with the wide range of functions played by E3 ubiquitin ligases[91], which include regulation of tissue-level phenotypes such as cardiac hypertrophy[92], here we report that WWP2 regulates IRF7 transcriptional activity in *Ccl5* expressing Ly6c[high] monocytes. This contributes to the pro-inflammatory and profibrotic macrophage activation and infiltration in the fibrotic heart. Consistently, the IRF7 regulon is mainly expressed within the Ly6c[high] monocyte cluster and IRF7 binds to Ccl5 promoter, regulating its transcription together with type I interferons in a WWP2-dependent manner. While dysfunctional WWP2 significantly reverts Ly6c[high] monocyte infiltration through IRF7 and Ccl5-mediated monocyte chemotaxis and activation, WWP2[Mut/Mut] macrophages showed significant upregulation of MAF regulon. MAF (also known as c-MAF) is a transcription factor that promotes M2-like macrophage polarization[93] and IL10 expression[94]. Thus, although the predominant anti-inflammatory and antifibrotic effect of WWP2 operates through IRF7-mediated Ccl5/Ly6c[high] monocyte axis, we cannot exclude a direct effect of WWP2 on homeostasis/repair macrophages through other immunomodulatory transcription factors (e.g., c-MAF) or via anti-inflammatory chemokines such as Ccl24.

As reported for other E3 ligases[95–97], WWP2 can have multiple targets in different cell-types. Previously we identified SMAD2 in fibroblasts[39]. In this study, we identify IRF7 as an interacting partner of WWP2, and suggest the mechanisms through which WWP2 exerts its M1-like polarization effects in cardiac macrophages. WWP2 promotes the (poly) mono-ubiquitination and transcriptional activity of IRF7. IRF7 is a transcription factor associated with M1-macrophage polarization[98] critical for LPS-induced type I IFN responses and IL-1β production in mice[99,100]. We report that WWP2 deletion causes down-regulation of IFN-γ, a type II interferon that has not been conventionally linked to IRF7. Interestingly, IRF7 has been described as a novel player in the IFN-γ response[101]. Previous studies showed how in vivo deficiency of IRF7 attenuates tissue fibrosis in mice[102], supporting regulation of cardiac fibrosis by WWP2 via IRF7. While we cannot rule out the contribution of IRF7 protein modifications other than ubiquitination[71], our data suggest a WWP2/IRF7 interaction, promoting the non-degradative ubiquitination of IRF7, and its subsequent phosphorylation and dimerization. This mechanism is in line with our previous report detailing how WWP2 mediates the TGFβ1-induced nucleocytoplasmic shuttling and transcriptional activity of SMAD2 in fibroblasts[39]. Since the interaction between IRF7 and WWP2 causes a similar transcriptional regulation in macrophages, the results presented here suggest cell-type specific targets of WWP2 in the regulation of fibrosis.

Dysfunctional WWP2 in ex vivo cultured macrophages causes reduced myofibroblast activation through soluble factors that remain to be identified. Here, we propose Ccl5 as a major Ly6c[high] monocyte-derived chemokine whose expression depends on functional WWP2. Although targeting Ccl5 is beneficial in hepatic fibrosis[103], it could be argued that Ccl5 effect is on the infiltration of monocytes.

Clinical trials testing anti-inflammatory strategies in cardiomyopathies had so far led to non-significant or even deleterious effects[104]. Global macrophage depletion[15,105] or TNF-α antagonism[106] during heart injury resulted in increased rates of mortality, suggesting that cellular depletion strategies must take into account the heterogeneity of cardiac macrophages. Thus, targeting specific macrophage subtypes rather than the entire population holds therapeutic potential in NICM. As such, we speculate that targeting WWP2 may have beneficial effects given its predominant regulatory effect on the IRF7-mediated Ccl5/Ly6c[high] monocyte axis, which unlike other monocyte/macrophage subtypes, has a driving role in promoting early fibrogenesis in NICM.

## Methods
### Mouse breeding
Mice were bred and maintained in animal facility at Duke-NUS medical school prior to use. All mice were housed in a specific pathogen–free (SPF) environment and their maintenance complied with all relevant ethical regulations according to guidelines issued by the National Advisory Committee on Laboratory Animal Research. The housing room was set to a 12 hrs light/dark cycle with lights off at 8 a.m., a temperature of about 22 °C, and a relative air humidity of ~50%. Protocol with IACUC number 2016/SHS/1170 was approved by Institutional Animal Care and Use Committee of the National University of Singapore, Duke-NUS Medical School.

### WWP2[Mut/Mut] mouse
We have previously generated the WWP2[Mut/wt] mouse line, in which a 4 bp deletion was generated after the stop codon of the mouse WWP2 allele[39]. WWP2[Mut/wt] mice were crossbred to generate WWP2[Mut/Mut] (Mut/Mut) and WWP2[wt/wt] (WT) mice in vivarium at Duke-NUS Medical School, Singapore. The background of mice is C57BL/6 J (B6J). Homozygous WWP2[Mut/Mut] and WT litters were weaned at around three weeks of age and housed in same-sex groups of four to five animals per cage for different experimental purposes.

### WWP2[flox/flox]LysM[Cre/wt] mouse
We generated WWP2[flox/flox] mice carrying a loxP pair in WWP2 by introducing the pair of loxP flanked exon 3 (Supplementary Fig. 15a). To delete the WWP2 gene in myeloid cells, WWP2[flox/flox] mice were bred with Lyz2[tm1(cre)Ifo/J] (also known as LysMcre, Stock No: 004781, Jackson Laboratory) mice in which the Cre recombinase is expressed under control of the murine lysozyme M gene regulatory region. The background of mice is C57BL/6J (B6J). WWP2[flox/flox]LysM[Cre] and WWP2[flox/flox]LysM[wt] litters were weaned at around three weeks of age and housed in same-sex groups of four to five animals per cage for different experimental purposes. A breeding scheme prevents embryonic deletion of WWP2 by Cre-expressing females. Female WWP2[flox/fWT]LysM[Cre] and male WWP2[flox/flox] mice were housed together in breeding cages. All cells in both the Cre-positive and Cre-negative mice maintain the *WWP2* gene on one allele.

### Cell cultures
**Macrophages.** BMDMs and SPMs were primarily cultured for in vitro experiments. In detail, bone cavity of the hind legs and the femurs obtained from the hip joint were flushed using a 10 ml syringe filled with DMEM attached to BD 26 ½" G needle (#21000578, BD PrecisionGlide). Spleen was extracted from the murine abdominal cavity and was crushed onto a 40 μm sterile filter with DMEM. After centrifuge, cell pellet was resuspended and cultured in macrophage medium (DMEM + 20% L929 Conditional Media +10% FBS + 1% Penicillin–Streptomycin) in 37 °C, 5% $CO_2$ incubator. On day 7, macrophages were obtained by non-enzymatic cell dissociation from the 75-cm² flask for experiments. Both BMDMs and SPMs were stimulated with LPS 100 ng/ml (#L3129, Sigma-Aldrich) + IFNγ 10 ng/ml (#575306,

Biolegend), or IL-4 10 ng/ml (#574304, Biolegend) + IL-13 10 ng/ml (#575902, Biolegend) or TGFβ1 5 ng/ml (#T7039, Sigma-Aldrich) for 4 hrs, 8 hrs, and 24 hrs, as indicated.

**Conditional medium.** BMDMs with a confluent cell number ($7.5 \times 10^5$/ well in six-well plate) were stimulated with LPS 100 ng/ml (#L3129, Sigma-Aldrich) + IFNγ 10 ng/ml (#575306, Biolegend). The conditional medium from treated and untreated cells was collected and stored for further analyses at −20 °C, after centrifugation.

**Cardiac (myo)fibroblasts.** Cardiac (myo)fibroblasts were obtained from the left ventricle (LV) of the mice heart as described previously[39]. Cardiac fibroblasts obtained (P0) were passaged, and only P1 or P2 cells were used in each experiment.

**NIH-3T3 cell line.** NIH-3T3 cell line was grown in DMEM medium supplemented with 10% fetal bovine serum. The cells were passaged twice before being used for experiments.

### Phenotyping and cell assays

**Angiotensin II-infusion model.** Angiotensin II (Ang-II) infusion model has been optimized and used in our previous study[39]. Briefly, Alzet miniosmotic pump (Model No. 1004, Durect Corporation) was subcutaneously implanted in eight-weeks old mice anaesthetized with 2% isoflurane. Miniosmotic pumps loaded with saline or Angiotensin II (#A9525, Sigma-Aldrich) were used to deliver Ang-II at 500 ng/kg/min. Mice were euthanized at the indicated time points, and hearts (left ventricle) were harvested for the experiments.

**Echocardiography analysis.** Transthoracic echocardiography was performed by one blinded specialist on echocardiography using Vevo 2100 (VisualSonics, VSI, Toronto, Canada) and a MS400 linear array transducer, 18- to 38-MHz under anesthetized condition. An average of 10 cardiac cycles of standard 2 dimensional (2D) were acquired and stored for subsequent analysis using Vevo Imaging Workstation version 1.7.2 (VisualSonics, VSI, Toronto, Canada). 2D guided M-mode of parasternal short-axis (middle) were selected for visualization of the papillary muscle during end systole and end-diastole. Left ventricular ejection fraction (EF) and fractional shortening (FS) were calculated using modified Quinone method, with the following formulas: LVEF = $(LVIDed^2 - LVIDes^2)/LVIDed^2$; FS = (LVIDed − LVIDes)/LVIDes; where LVIDed is left ventricular internal diameter at end-diastole and LVIDes is left ventricular internal diameter at end systole.

**Flow cytometry analysis.** Mouse heart (left ventricle) was perfused[107], excised, minced, digested with Collagenase II (#LS004174, Worthington Biochemical Corporation) and Dispase II (#494207800, Roche). LV tissue mixture was then mechanically disrupted and filtered through 70-μm cell strainer to obtain single-cell suspensions. After blocking with CD16/32 (#14-0161-85, Thermo Fisher Scientific, 1:100) at RT for 10 min, cells were collected by centrifugation, subjected to live-death dye and surface antibody staining, and analyzed or sorted by flow cytometry using BD FACS ARIA system, and the data were analyzed using FlowJo v10. Cardiac macrophages, cardiac monocytes, or Ly6C$^{high}$ monocytes were gated from whole heart cells as CD45+ CD64+ CD11b+ F4/80+ Ly6G− or CD45+ CD64+ CD11b+ or CD45+ CD64+ CD11b+ Ly6C$^{high}$ cells, respectively. The specific antibodies used for flow cytometry are listed as follows: Anti-Ly-6C Monoclonal Antibody PerCP-Cyanine5.5 (#45-5932-82, Thermo Fisher Scientific, 1:25), APC Rat anti-mouse Anti-CD11b (#561690, BD Biosciences, 1:100), FITC Mouse Anti-Mouse CD45.2 (#561874, BD Biosciences, 1:100), APC-H7 Rat Anti-Mouse Ly-6G (#565369, BD Biosciences, 1:200), PE Mouse anti-Mouse CD64 (#558455, BD Biosciences, 1:100), BV421 Rat Anti-Mouse F4/80 (#565411, BD Biosciences, 1:100), mouse anti-mouse CD45 (#561874, BD Biosciences, 1:100), rat anti-mouse IRF7-PE (#12-5829-82, Biolegend, 1:100).

**Nuclear translocation assay.** BMDMs were stimulated with LPS 100 ng/ml (#L3129, Sigma-Aldrich). After fixation and permeabilization with eBiosienceTM Intracellular Fixation and Permeabilization Buffer Set (#88-8824-00, Life Technologies Holdings), intracellular IRF7 was detected using an IRF7 antibody (#GTX01065, GeneTex, 1:100) followed by Goat anti-Rabbit IgG (H + L) Cross-Adsorbed Secondary Antibody 488 (#A11008, Thermo Fisher Scientific). DRAQ5 (#564903, BD Pharmingen) was used as nuclear dye. Images were acquired using Amnis ImageStream X Mk II imaging Flow cytometer using ×40 objective; nuclear translocation was quantified using the Similarity Score feature within the IDEAS software package[108].

**Chromatin immunoprecipitation-qPCR assay.** Chromatin immunoprecipitation (ChIP) assay was performed with anti-IRF7 antibody (#ABF 130, Sigma-Aldrich). Briefly, BMDMs were grown for 90% confluence. DNA and proteins in the cells (-$10^6$ each sample) were cross-linked using 1% (v/v) formaldehyde and then sonicated in lysis buffer to obtain 200 bp–500 bp long DNA fragments. The lysates were either used for IRF7 immunoprecipitation or kept as input control. Reverse-crosslinking and elution of DNA was followed by immunoprecipitation with antibodies. For the ChIP-seq data analysis, IRF7 binding site motif matches were found at *Ccl5* and *Irf7* loci using the FIMO algorithm[109]. ChIP-qPCR was performed with primers specific to the binding sites (described in Supplementary Fig. 17, and listed in Supplementary Table 1). Fold of enrichment values were determined according to the ratio $2^{-\Delta\Delta CT}$ where ΔCT = $Ct_{IRF7}$-$Ct_{INPUT}$. Ratios were determined from ChIP experiments with at least three independent BMDM cell isolates, and each qPCR assay was performed in triplicate.

**Histology and immunofluorescence staining.** 5 μm thick LV sections were fixed in 10% Neutral Buffered Formalin and processed using Leica automatic tissue processor. Following dewaxing and rehydration, the LV sections were stained with Sirius Red staining kit (#9046, Chondrex, Inc.) and Masson's Trichrome Staining as per manufacturer's instructions. Primary cultured cells or sorted macrophages were seeded on 8 well removable silicone chamber slides (#80841, ibidi) or a μ-Dish 35 mm (#81176, ibidi) for co-culturing. Chambers or dishes were fixed in ice-cold acetone for 30 min at room temperature. Antibodies used for immunofluorescent staining are as follows: anti-Col1a1 (#ab6308, abcam, 1:100), anti-CD68 (#ab125212, abcam, 1:100), anti-ACTA2 (#A5228, Sigma-Aldrich, 1:100), anti-CD45 (#561874, BD Biosciences, 1:100), anti-CCR2 (#MAB55382-100, RnD Systems, 1:100), anti-EGR1 (#sc515830, Santa Cruz Biotechnology, 1:100), anti-CD74 (#NBP2-29465, Novus Biologicals, 1:100), anti-Lyve1 (#14-0443-82, Life technologies holdings, 1:50), anti-IRF7-PE (#12-5829-82, Bio-legend, 1:100), and anti-WWP2 (#A302-936A, Bethyl Laboratories, 1:100). Images were visualized using Secondary Antibodies (A11008, A11001, A11011, A11004 and A11077, Thermo Fisher Scientific) at 1:500 for 2 h at room temperature. DAPI (#D1306, Thermo Fisher Scientific) and Alexa Fluor 647 Conjugate Wheat Germ Agglutinin dye (#W32466, Life technologies holdings) were used to visualize the nuclei and membrane of myocytes, respectively. Sections were imaged under a Leica Fluorescent microscope and ZEISS 720 confocal Laser Scanning Microscope.

**Immunofluorescence analysis.** ImageJ 1.53k software was used to analyze the fluorescent microscopy images. For analysis using ACTA and Col1 staining[110], the fluorescence intensity (MFI) was quantified after choosing a field as a region of interest (ROI), while a non-florescent area of the same image was measured as background. Cell

counting from the same field was automatically assessed with DAPI staining. The intensity of positive cells is calculated as $(MFI_{ROI}-MFI_{background})$/cell number in the field. For colocalization of WWP2 and IRF7 signals, the intensity profiles of WWP2 and IRF7 were recorded crossing the nuclei of representative cells. The MOC and Pearson's correlation coefficient were calculated according to the methods described previously[68].

**Cardiomyocyte size.** Rhodamine wheat germ agglutinin (WGA, Vector laboratories, #RL-1022) was used to stain the cardiomyocytes. Cardiomyocyte cross-sectional area was determined on digitized images of WGA-staining; such images were obtained using a fluorescence microscope (Olympus BX61) attached to a digital camera and connected to a computer equipped with image analysis software. Cardiomyocytes' outlines were traced and cell areas measured using Image-Pro Plus 6.0 (Media Cybernetics, Inc., Silver Spring, Md). Four field images were averaged for a single mouse; each field image included at least 80 cross-sections of myocytes.

**Hydroxyproline assay.** The amount of total collagen in the LV was quantified using the Quickzyme Total Collagen assay kit (#QZBtot-col1, Quickzyme Biosciences). The assays were performed according to the manufacturer's protocol.

**siRNA experiments.** For siRNA transfection, NIH-3T3 cells were seeded on a 96-well plate (~70%) and were transiently transfected with siRNA duplexes (20 nM) designed for targeting 5' or 3' in *Wwp2* mRNA (Qiagen) using Lipofectamine RNAiMAX (#13778075, Life technologies holdings) in a serum free medium for 48–72 hrs according to the manufacturer's instructions.

**Luciferase assay.** Cells were transfected with ISRE luciferase reporter vector and non-inducible firefly luciferase vector premixed with constitutively expressing Renilla luciferase vector as a negative control (#60613, BPS Bioscience). 24 hrs after transfection, cells were treated with vehicle or 100 ng/ml LPS with 10 ng/ml IFNγ for 4 hrs and harvested. Luciferase assays were performed using the Dual-Luciferase Reporter Assay System (#E1960, Promega).

**ELISA assay.** Cultured BMDMs were seeded in a six-well plate ($7.5 \times 10^5$/well). After stimulation with LPS 100 ng/ml (#L3129, Sigma-Aldrich) + IFNγ 10 ng/ml (#575306, Bio-legend), the conditional medium from treated and untreated cells was collected and stored for further analyses at −20 °C, after centrifugation. The levels of IL-6 and CCL5 in equal volumes of cell culture supernatant were quantified using IL-6 Mouse ELISA Kit (#BMS603-2, Invitrogen) and Mouse RANTES (CCL5) ELISA kit (#88-56009-22, Invitrogen) following manufacturer's instructions.

**Western blot assay.** Cell lysates were obtained from cardiac tissues and cells using RIPA buffer (#89900, Thermo Fisher Scientific) supplemented with proteases (#11836170001, Sigma-Aldrich) and phosphatase inhibitor cocktail (#PHOSS-RO, ROCHE). For native polyacrylamide gel electrophoresis (PAGE), cell lysates were obtained using NP-40 lysis buffer (#FNN0021, Life Technologies Holdings) supplemented with protease inhibitors (#P2714, Sigma- Aldrich) and PMSF (#10837091001, Sigma-Aldrich) following manufacturer's instructions. Nuclear extracts were obtained from the BMDMs treated with LPS (100 ng/ml) + IFNγ (10 ng/ml) for 4 hrs as per the manufacturer's instructions using Pierce Direct NE-PER kit (#78833, Thermo Fisher Scientific).

**Co-immunoprecipitation.** The lysates were obtained from the BMDMs (~5 × 10⁶ in each sample) and subjected to co-immunoprecipitation as per the manufacturer's instructions using Pierce Direct Magnetic IP/CO-IP kit (#88828, Thermo Fisher Scientific).

**Ubiquitination analysis.** Cells were subjected to treatment with proteasomal inhibitor MG132 (#M7449, Sigma-Aldrich) for 30 min prior to treatment with LPS (100 ng/ml) + IFNγ (10 ng/ml) for 4 hrs. Immunoprecipitates were washed from conjugated beads and boiled in 5× sample buffer.

**Gel electrophoresis.** Lysates were routinely subjected to sodium dodecyl-sulfate (SDS) electrophoresis using a 4–12% polyacrylamide gel after Breadford quantification. A 10% Native gel was pre-run in 25 mM Tris/192 mM glycine, pH 8.3 supplemented with 1% Sodium Deoxycholate (# 89904, Life Technologies Holdings) for 30 min at 0°C. Cell lysates were mixed with Native page sample buffer (#1610738, BIORAD). Electrophoresis was carried out at 25 mA for 60 min at 0°C or until the dye reaches the bottom. The gel was incubated with 25 mM Tris/192 mM glycine, pH 8.3 supplemented with 0.1% SDS for 30 min at room temperature before transferring onto a polyvinyldiene difluoride membrane for 45 min at 90 V at 4°C.

**Band detection and visualization.** Blotting of the membrane was performed using anti-WWP2 (#A302-936A, Bethyl Laboratories, 1:500), anti-ACTA2 (#A5228, Sigma-Aldrich, 1:10,000), anti-Vimentin (#ab45939, Abcam, 1:500), anti-Periostin (#NBP1-30042, Novus Bio, 1:500), anti-Fibronectin (#SAB4500974, Sigma-Aldrich, 1:500), anti-Ubiquitin (#3933, CST, 1:500), IRF7 (#GTX01065, GeneTex, 1:500), p-IRF7 (#24129 S, CST, 1:500), p-IRF7 (#PA564834, Thermo Fisher Scientific, 1:500), IFITM3 (#PA511274, Thermo Fisher Scientific, 1:500), S100A8 (#ab180735, Abcam, 1:500), IFNβ (#ab218229, Abcam, 1:500), IFNγ (#ab133566, Abcam, 1:500) CCL2 (#ab25124, Abcam, 1:500), CCL5 (#sc-365826, SantaCruz, 1:500), iNOS (#ab15323, Abcam, 1:500), and IL-6 (#ab208113, Abcam, 1:500) after transfer onto a nitrocellulose membrane at 100 V for 1 hr. Blots were visualized with anti-Rabbit HRP (#A120-101P, Bethyl Laboratories, 1:5000) and anti-Mouse HRP (Bethyl laboratories, #A90-116P, 1:5000). Protein A-HRP (#101023, Thermo Fisher Scientific, 1:1000) on a Kodak automated developer after blocking with 5% non-fat dry milk for 1 hr at room temperature using Pierce ECL Chemiluminescent substrate (#32106, Thermo Fisher Scientific). Immobilon Forte Western HRP substrate (#WBLUF0500, Merck), and quantified using densitometry using ImageJ. Anti-Tubulin (#T5168, Sigma-Aldrich, 1:5000) and anti-GAPDH (#ab8245, Abcam, 1:5000) were used as loading controls. Anti-Lamin A/C (#ab8984, Abcam, 1:5000) was used as nuclear control. Full and unprocessed scanned images of the blots are provided in the Source Data file.

**qRT-PCR analysis.** Total RNA was extracted from snap-frozen fibrotic cardiac tissue, sorted cardiac macrophages (CD45⁺CD64⁺CD11b⁺F4/80⁺Ly6G⁻) and derived macrophages using the RNeasy mini kit (#74106, Qiagen), and cDNA was prepared using iScript cDNA synthesis kit (#170-8897, BIO-RAD) according to the manufacturer's instructions. Fast SYBR-Green master mix (#170-8880AP, BIO-RAD) was used for the analysis of gene expression using the BIO-RAD CFX RT-PCR system. The primers used in the experiment are listed in Supplementary Table 1. 18 S was used to normalize the relative gene expression and the $2^{-\Delta\Delta Ct}$ method was used to measure fold changes in gene expression.

## Transcriptomics analysis

**Single-cell RNA-seq protocol.** Isolated single-cell suspensions were converted to barcoded scRNA-seq libraries by using the Chromium Single-Cell 3' Library, Gel Bead & Multiplex Kit, and Chip Kit V3, loading an estimated 7000–12,000 cells per library/well, and following the manufacturer's instructions. Indexed libraries were sequenced using Illumina HiSeq 4000 sequencer, where 150 bp pair-end sequences

were obtained. Sequencing reads were aligned and quantified to the mouse genome GRCm38 (mm10-3.0.0 provided by 10x Genomics) using 10x Genomics cellranger count (v3.1.0).

**Single-cell RNA-seq analysis.** For scRNA-seq analysis (at 7 days after Ang-II-infusion), pre-processing involved removing genes that are not detected in at least 10 cells. Cell filtering criteria include filtering of cells with number of features and/or number of counts outside the 10th and 90th percentiles. For cell-type annotation, markers from Skelly et al[30]. were utilized. For each library, we employed Seurat (v4.0.1)[111] for normalization using the "LogNormalize" method and a scale factor of 10,000. FindVariablesFeatures() function was used to obtain the top 5000 feature genes. Mitochondrial genes were removed. ScaleData() function was used for scaling, while dimension reduction was accomplished using RunPCA() and RunUMAP() functions with number of dimensions set at 50. The Scds (1.6.0) package[112] was used to calculate doublet hybrid score; cells with hybrid score >0.5 were considered doublets and removed. Cell cycle scoring was done using the CellCycleScoring() function with markers derived from[113]. After QC and filtering, the 4 libraries (WT, WT Ang-II, WWP2^Mut/Mut, and WWP2^Mut/Mut Ang-II) are subsequently merged, yielding an expression matrix of 16,273 genes and 13,014 cells.

For downstream analysis of immune cells, each immune cell-type is subset from the total expression matrix and subjected to the same Seurat's pipeline described above. Clustering was performed using Seurat's FindNeighbors() and FindClusters() functions with a resolution of 0.8 and 50 Principal Component Analysis (PCA) dimensions. Violin plots were plotted using Seurat's VlnPlot() function. With respect to the analysis of the cardiac macrophage cell compartment, two macrophage clusters contained very few cells (<100) and expressed markers for B cells and neutrophils (see Supplementary Fig. 2b–e for details). These could represent macrophages that are enriched for B-cell/neutrophil markers (or possibly doublets), and were filtered out in our downstream analyses. Ccr2+ macrophages were defined as macrophages whose scaled expression of *Ccr2* is higher than the 70th percentile with respect to the distribution of *Ccr2* scaled expression in macrophages (see Source Data for more details).

**Differential expression analysis.** For pairwise differential expression analyses (e.g., WWP2^Mut/Mut Ang-II vs WT Ang-II, etc.), the Seurat's FindMarkers() function was used to obtain the DEGs with false discovery rate < 5%. For determining cluster markers or cell-type markers, the Seurat's FindAllMarkers() function was employed and the Wilcoxon rank sum test was used. For the derivation of cell-type markers, we used the following parameters: min.pct = 0.25 and logfc.threshold = 0.25. For the derivation of macrophage subcluster markers, we used the following parameters: min.pct = 0.1 and logfc.threshold=0.5.

**Pseudotime analysis.** Monocle3 (v1.0.0)[48] and Slingshot (v1.8.0)[50] were used for calculating pseudotime from the scRNA-seq data. Seurat-Wrappers (v0.3.0) was used to convert Seurat objects into cell dataset objects. Two outlier clusters S1 and S2 (see Supplementary Fig. 2b–e for details) were removed prior to calculation of pseudotime. Ordering of cells was performed using the uniform manifold approximation and projection (UMAP) dimension reduction; in this analysis cluster 5 was chosen as the root cell cluster.

**Functional enrichment analysis.** GSEA, and therein the clusterProfilier's (v3.18.1)[114–116] GSEA() function was used, using functional gene sets downloaded from Bader lab[117]. The normalized enrichment scores (NES) were weighted by the overlap between the input gene sets and the canonical functional gene sets. The Benjamini-Hochberg (BH) method was used for multiple testing correction, and the BH adjusted *p* value cutoff was set at 0.05 unless otherwise indicated. In addition, the hypergeometric test within the enrichGO() function of

clusterProfiler (version 3.0.4) was used with functional gene sets downloaded from Bader lab, followed by BH multiple testing correction. The simplify method of clusterProfiler was used to filter out semantically redundant gene ontology (GO) terms with the cutoff set at 0.5, select_fun = "min", and by = "p.adjust". For all cluster profiler analyses, bitr() was used to convert gene symbols to EntrezID using either the databases org.Hs.eg.db (v3.12.1) or org.Mm.eg.db (v3.12.1) as appropriate. For Reactome pathway functional enrichment analysis of the WWP2-regulated Irf7 target genes, we used the STRING database v11.5 (https://string-db.org/, accessed on 15th December 2021) with default settings but removing "text mining" as an interaction source.

**Ligand-receptor interaction analysis.** CellPhoneDB (v2.0) was employed for ligand-receptor interaction analysis using default settings. Two sets of analyses were performed: (1) in Wwp2^−/− Ang-II *vs* WT Ang-II *vs* WT, and (2) in all 7 macrophage/monocyte clusters regardless of genotype. NicheNet (1.0.0)[118] was used to convert mouse gene symbols to human analogs. Because CellPhoneDB database of ligand-receptor interaction nomenclature includes both "ligand-receptor" and "receptor-ligand" notations, all derived interactions were standardized to follow only the "ligand-receptor" notation. Only chemokine ligand-receptor (CCL-, CCR-, and CX-family) interaction pairs were considered. The Sankey flow diagram was constructed using the plotly (v4.9.3) package in R. Differential expression analysis of respective ligands and receptors was performed using a two-sided Wilcoxon test followed by BH correction for multiple testing. It is important to note that mice-human orthologues of *Ccl12/CCL2* and *Ccl2/CCL13* were used because CellPhoneDB only recognizes human genes. For differential expression analysis of ligand-receptor pairs, we calculated the mean expression of each ligand-receptor pair in WWP2^−/− and WT macrophages separately, and then employed a two-sided Wilcox followed by BH correction for multiple testing. The difference in expression was quantified as the $\log_2$(fold change) of the mean expression calculated across WWP2^−/− macrophages over the mean expression calculated across WT macrophages.

**Regulon analysis.** We derived transcriptional networks (regulons) for the mouse cardiac macrophages by pyScenic standard pipeline (pySCENIC v0.11.2)[119] with default parameters, and random seed = 10. Briefly, pySCENIC (1) derives a set of gene co-expression networks defined by a transcription factor (TF) and its target genes, (2) evaluates a network for enrichment of TF-specific *cis*-regulatory elements and removes targets genes lacking an enrichment of these elements, and (3) assesses the activity level of the network in each individual cell by the "Area Under the Curve Score" (AUC). In step 1 (network construction), we used the raw RNA counts of the macrophage transcriptome (1,765 cells by 14,847 genes) as input, and a list of TFs specific to the murine genome (mm10) from https://github.com/aertslab/scenic-nf/blob/master/example/allTFs_mm.txt as the TF candidates for the network analysis. For each expressed gene in the transcriptome, a tree-based regression model was built with the TF candidates as predictors using the GRNBoost2 algorithm. The regulatory relationship between a TF and a gene was determined by the weight of the TF in the regression model. In step 2 (network refinement), we used the following databases of genome-wide regulatory features ("mm10_refseq-r80_500bp_up_and_100bp_down_tss.mc9nr.feather") and TF motifs ("motifs-v9-nr.mgi-m0.001-o0.0.tbl ") to assess a network for the enrichment of regulatory features and prune its members. In brief, the database contains pre-computed rankings of genome-wide regulatory features specific to a TF, and using these rankings pySCENIC calculates the enrichment of these features in the targets genes. In step 3 (the evaluation of regulon activity), pySCENIC ranks each gene in the transcriptome of a cell by expression, and an AUC score evaluates

the enrichment of the members in a regulon based on this ranking. This approach was used to identify regulons (TFs and the network of the respective downstream targets) for our cardiac macrophage dataset consisting of 1,765 macrophages (WT, WT Ang-II, Wwp2$^{-/-}$, and Wwp2$^{-/-}$ Ang-II). In total, 353 regulons were identified. In order to ascertain the regulons most differentially regulated by Wwp2$^{-/-}$, we first quantified the expression of each of the 353 regulons in both WT Ang-II and Wwp2$^{-/-}$ Ang-II macrophages by Seurat's AddModuleScore() function. A two-sided Wilcoxon test was employed to determine the differential expression of each regulon with respect to Wwp2 dysfunction (i.e., Wwp2$^{-/-}$ Ang-II vs WT Ang-II). P-values were adjusted for multiple testing using the BH method. We further filtered down the list of differentially regulated regulons using the following criteria: (1) the average expression of the regulon must be greater than 0, (2) the regulon must contain at least 1 DEG (Wwp2$^{-/-}$ Ang-II vs WT Ang-II), and (3) the proportion of DEGs (Wwp2$^{-/-}$ Ang-II vs WT Ang-II) must be at least 5% of the total regulon size. This yielded a total of 34 significantly differentially regulated (BH corrected $P < 0.05$) and expressed regulons which also contain TF-target genes regulated by Wwp2.

**Comparison with published datasets.** The M1 and M2 macrophage gene sets were derived from Orecchioni et al.[120] who compared in vivo and in vitro M1 and M2 signatures. For the gene set associated with IRF7 binding, the GTRD[121] was used (https://gtrd.biouml.org/). IRF7 peaks were obtained for myeloid cells with max gene distance set at 5000 base pairs. The final gene set was filtered using the criterion of at least 1 site count. BiomaRt (v.2.46.3) (www.ensembl.org)[122] was used to convert human gene symbols to mice gene symbols unless otherwise indicated. Comparison of our cardiac macrophage dataset and clusters with transcriptomic datasets of murine myocardial models was performed by calculating the Spearman correlation of specificity matrices of respective datasets. This was derived from the "gene-specificity index" employed by Tosches et al.[123], which was used to calculate correlation matrices of expression data across different species[124]. In brief, each pairwise comparison is done in three steps. First, a common gene set is derived between the two datasets. Second, the specificity of each gene is quantified across the clusters (which could be a priori cell types or data-driven clusters depending on the inquiry); this is obtained by using the ratio between gene expression in the cluster of interest and the mean gene expression in the entire dataset. The specificity index varies between 0 and N, where N is the number of clusters present in the dataset. A specificity index of 1 indicates that the gene is equally expressed in all clusters while a specificity index of N indicates that the gene is only expressed in the cluster of inquiry. Third, the pairwise Spearman correlation was quantified with respect to the specificity matrices of the two datasets. For the comparison with Dick et al.[33](myocardial infarction (MI) model, MI subset by scRNA-seq), the common gene set was derived by overlapping the top 5000 feature genes (which was calculated by Seurat's FindVariableFeatures() function in both their and our WT macrophage dataset). Note that the comparison was made only with respect to the main clusters derived from the Dick et al. analysis ("Timd4 cluster", "MHC-II cluster", "Ccr2 cluster", "Isg cluster", "Proliferating Macrophages", and "Monocytes"). For Wong et al.[51](microarray), the common gene set was derived by overlapping the top 5000 feature genes of our WT macrophage dataset and all genes detected in the Wong et al. microarray dataset. Note that comparison was made only with respect to the clusters ("Monocytes", "Ccr2+", "Ccr2-MHC+Flt3−", and "Ccr2− MHC− Flt3−"). Furthermore, a resident macrophage signature was derived in a murine model of Ang-II-infusion from Zaman et al.[38]. For comparison with Ni et al. (TAC model)[52] we looked at the normalized expression of *Cd72* in our dataset, which was plotted across the monocle-derived pseudotime trajectory derived in WT and WT Ang-II macrophages in our study. The cosine similarity was used to quantify similarity of *Ccr2* and *Cd72* normalized expression trajectories with P value derived from 1000 random permutations using cosinePerm() function from the PharmacoGx package (v2.2.4)[125].

**Statistical analyses.** Data are expressed as mean ± standard deviation (SD). The applied statistical tests, which were dependent on the number of groups being compared and the study design, are indicated in each figure legend. Unless otherwise indicated, a two-tailed non-parametric Mann-Whitney U test was used to compare two groups, with * denoting $P < 0.05$, and ** denoting $P < 0.01$. All experiments requiring the use of animals, directly or as a source of cells, were subjected to randomization. The experimenters were blinded to the grouping information. All in vitro experiments were independently replicated at least three times as indicated in the figure legends.

In addition to the specific R (v4.0.4) packages used in specialized analyses, other general use packages include: readxl (v1.3.1), ggplot2(v3.3.5)[126], scran (v1.18.7)[127], rColorBrewer (v1.1.2), ggrepel (v0.9.1), pheatmap (v1.0.12), dplyr (v1.0.7), ggpubr (v0.4.0), reshape2 (v1.4.4), tidyverse (v1.3.1)[6], VennDiagram (v1.7.1)[128], magrittr (v2.0.2), Biobase (v2.50.0), stringr(v1.4.0), psych (v2.1.9)[129], ggbeeswarm (v0.6.0), ggridges (v0.5.3), Matrix (v1.4.0), ggnewscale (v0.4.5), SeuratWrappers (v0.3.0), gridExtra (v2.3), data.table (v1.14.2), Binarize (v1.3), lsa (v0.73.2), AnnotationDbi(v1.52.0), CoreGx (v1.2.0)[125], DOSE(v3.16.0)[130], slingshot (v1.8.0)[50] and matrixStats (v0.61.0).

### Reporting summary

Further information on research design is available in the Nature Portfolio Reporting Summary linked to this article.

## Data availability

All the data generated in this study supporting the main findings have been deposited to NCBI's Gene Expression Omnibus (GEO) and are accessible via GEO Series accession number GSE198003. The processed data are comprised in Source data and provided as a Source Data file, representing numerical data shown on figures. Relevant raw data are included in Source data and provided with this paper. Source data are provided with this paper.

## Code availability

We used published and publicly available algorithms. See method sections for full description of each analysis including input data, library and algorithm version used, for which we provide corresponding web links.

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

## Acknowledgements

The research was primarily supported by the Academic Research Council, Ministry of Education (ARC, MOE) for AcRF Tier 2 funding (T2EP30221-0013) to E.P. and National Natural Science Foundation of China (No. 81830020) to A.H. We acknowledge additional funding support from Duke-NUS Medical School to E.P. and J.B.

## Author contributions

E.P. and H.C. conceived and managed the study, obtained and managed funding for the project. Mouse animal and cell experiments were carried out by H.C., with the assistance of N.D. and J.Z.L. G.C. conceived and carried out Systems Genetics and Bioinformatics analyses with the assistance of K.Y.-H., J.G. and S.L. E.L.S.T. managed the budget and administrative work throughout the project. N.G.Z.T. carried out the mice heart echocardiography measurements. M.M.-M. and M.K.-S. provided the LysM^cre mice strain. S.C. and A.H.-Z. generated the WWP2^flox/flox mice. H.C., E.P., and J.B. designed all analyses, experiments and wrote the manuscript with inputs from co-authors. H.C. and G.C. contributed equally to this work.

## Competing interests

The authors declare no competing interests.
