## [Peer Review File · Nature Communications]

The E3 ubiquitin ligase WWP2 regulates pro-fibrogenic monocyte infiltration and activity in heart fibrosisREVIEWER COMMENTS

Reviewer #1 (Remarks to the Author):

Chen et al. present a work on the role of E3 ligase WWP2 in myeloid cells in mouse model of Ang II cardiomyopathy. This is a follow up study of previously published data on WWP2mut mice in the same model. In the current study, authors analyze in details phenotype of myeloid cells not only in WWP2mut mice but also in LysM-Cre x WWP-flox (newly generated). Authors find that expression of WWP2 in myeloid cells control accumulation of profibrotic macrophages and Ly6c(hi) monocytes in the heart and thus cardiac fibrosis and heart failure in Ang II model. On molecular level WWP2 regulates IRF7 pathway and monocyte polarization. Great part of the data is based on scRNAseq analysis.

The concept of macrophage involvement in heart failure is well known and this work provides a novel molecular insight. In general, molecular part was performed carefully providing high quality data. On the other hand, there are certain open questions mostly related to pathophysiology of the model.

Major concerns:

- in both transgenic mice, there is reduced influx of monocytes and macrophages at d7. Experiments do not explain this observations but it seems that regardless of molecular signature, the number of cardiac monocytes/macrophages at d7 is responsible for the phenotype at d28. Does WWP2 in myeloid cells affect their migration, proliferation, survival etc? Is there a difference in these cells (in the heart and in periphery) in normotensive mice?
- What about other cardiac infiltrates (T, B cells, neutrophils etc.) in WWP2Mac mice?
- the study is performed on a model of cardiomyopathy but no data on cardiomyopathy and cardiomyocyte hypertrophy is given
- WWP2 mRNA and protein levels should be shown for cardiac macrophages sorted from WWP2Mac mice at d7 (see Fig. 4)

Reviewer #2 (Remarks to the Author):

In the manuscript by Chen, et. al., the authors address the role of WWP2, an E3 ubiquitin ligase, in the progression of cardiac fibrosis. The authors utilize whole body WWP2-mutant and macrophage (LysMcre) conditional WWP2-flox deletion models to address the role of WWP2 in monocyte/macrophage lineage. They use scRNA-seq to describe changes in myeloid cell gene expression following ANGII-mediated cardiac inflammation, including gene expression programs associating CCL5 and IRF7 pathways to loss of WWP2 in monocyte and macrophage populations. A variety of biochemical assays were performed both ex vivo and in vitro, the authors show a mechanism by which WWP2 binds directly to IRF7 in myeloid cells to regulate its mono-ubiquitination. In the absence of WWP2, there is reduced IRF7 present in the cell, which is needed for the expression of CCL5 through direct chromatin interaction. This pathway leads to attenuation of CCR2+ classical monocyte recruitment and reduction in the formation of fibrosis in knockout mice.

Overall this manuscript uncovers a novel connection between WWP2 and its effect on myeloid cells. Mechanistic assays are a major strength of the paper and convincingly link the proposed connection between WWP2-IRF7-CCL5. However, there remains a few lingering questions that would help strengthen the in vivo mechanism and the relationship of WWP2 in immune and non-immune cells.

Major:

- 1) What is the steady state (or non-inflamed) effect of LysMcre WWP2-flox on cardiac macrophages and circulating monocytes and neutrophils? It would be helpful to know whether local or systemic defects are present prior to the ANGII infusion experiments. Please report macrophage data as total cells/heart and percentage of CD45+, and blood cells should be presented as cells/mL blood. In keeping with this data reporting approach, could the authors also provide immune cell counts for the hearts in the ANGII experiments that are presented? The flow plots presented in Figure 2 are difficult to interpret because even non-macrophages appear to be

dramatically reduced (such as CD11b+ F4/80-), and the negative populations appear to be very different autofluorescence. This makes the data presented as a % of total cells difficult to fully interpret. From experience, knowing total macrophage counts/heart and percentage of CD45+ helps to draw conclusions regarding expansion or lack-there-of in tissues. Finally, did the investigators perform any i.v. labeling controls to determine whether all subsets of immune cells are indeed infiltrating into the heart or merely accumulating along the vasculature?

2) The authors describe defects in monocyte recruitment in the whole body WWP2^{-/-} and conditional knockout systems but do not perform monocyte recruitment assays. Alternative possibilities to defective recruitment could be that WWP2 monocytes are dying within the tissue, or potentially re-circulating following entry. Could authors provide supporting data for their conclusion? Similarly, monocle data predicting monocyte differentiation paths following entry into the heart is presented in a somewhat unconventional fashion. Could the authors present this data on a UMAP projection? Based on the limited data shown, it also appears like cluster 6 and cluster 5 do not appropriately fit in the trajectory well and might be more suitable to be excluded. In addition, it has become increasingly common to provide multiple trajectory analysis approaches for scRNA-seq datasets. Adding slingshot or RNA-velocity analysis might help strengthen arguments regarding the interconnectedness between clusters and predicted differentiation pathways.

3) Since there is reduced monocyte recruitment, do tissue resident cardiac macrophages proliferate at a faster rate in the WWP2^{-/-} mice, compared to controls in the steady state or following ANGII?

4) The authors show that conditional depletion of WWP2 in macrophages regulates the fibrotic response to ANGII infusion. The magnitude of protection is equal to the total protection previously found in the total body knockout (Chen, Nature Communication 2019). Are the authors able to weigh the differential roles of WWP2 in cardiac fibroblasts against macrophages? Experimentally, it would be valuable to know if replacement of bone marrow in WWP2-mutant mice with WT donor cells (thus creating WT-macrophages but KO stroma) is sufficient for protection (or loss) from the fibrotic response. In a similar line of experiments, performing mixed bone marrow chimera experiments (50% WT, 50% KO) comparing cell intrinsic versus extrinsic influences of WWP2-deletion on monocytes and their ability to be recruited into the heart may prove to be valuable to the interpretation of this project.

Minor:

Page 14, Line 419: I believe Figure 5J is incorrectly referenced when the authors are referencing Fig 6D.

Method description for monocle appears to have description errors in terms of excluded data clusters and start point for pseudotime.

How were cardiac macrophages sorted for qPCR analysis? Were monocyte excluded?

Figure 3J – please describe how fluorescence images were normalized and intensity was calculated.

A recent and potentially relevant citation for immune mechanisms of cardiac fibrosis should be included, Revelo et. al. Circulation Research 2021

Page 35 line 17-19: Statement needs revised from how it currently reads; I believe the authors are referencing the breeder pairs only.

Reviewer #3 (Remarks to the Author):

In this paper, Chen et al described a role of the E3 ligase WWP2 in heart fibrosis via regulating a Ly6C-high subpopulation of monocytes. By using single cell profiling, they found that Ly6C-high monocytes were increased upon angiotensin II treatment, which were reversed in WWP2-ablated mice. WWP2-deficient monocytes showed reduced expression of several chemokines like CCL5, and other pro-inflammatory cytokines such as IL6, TNF α , and IL1 β , but increased expression of IL4, which was further confirmed by conditional deletion of WWP2 in monocytes. Transcriptional profiling suggested that IRF7 served as a target which was decreased in WWP2-deficient monocytes, and could drive the expression of itself (IRF7) and CCL5. They further provided results

showing that WWP2 interacted with IRF7, promoted its ubiquitination and nuclear translocation. The results seem to agree with their previous publication showing that WWP2 controls heart fibrosis via regulating the ubiquitination of Smad2 in cardiac fibroblasts (ref. 37, NC 2019). However, it is concerned that the key data are preliminary and largely correlative, and therefore not entirely convincing at the current stage.

Major points:

1, The observed reduction of Ly6C-high monocytes in WWP2-deficient mice (Fig. 1) suggested that WWP2 may function in an early phase of inflammation via a direct effect on the production of pro-inflammatory cytokines such as IL-6, TNF α , and IL-1 (Fig. 3 and Fig. S10 to 11), rather than an indirect effect on CCL5-mediated infiltration. The authors should carefully examine the roles of these cytokines in their model.

2, Similarly, the authors claimed that CCL5 was a critical mediator for their observed phenotypes (Figs. 2 and 4), but they did not provide direct evidence to support such claims in both in vitro and in vivo systems.

3, It is generally believed that the M1 to M2 transition of macrophages, as they observed in WWP2-deficient mice and in vitro culture system (Fig. 3, and Fig. S9 to 12), will promote late-stage fibrosis/repair response. This observation seems to contradict with the authors' claims.

4, Another concern is from the authors' observation that WWP2 deficiency significantly reduced the numbers of Ly6C-high monocytes (Fig. 2C to E and 4G to I), which should be experimentally tested to see whether such cell number reduction is IRF7- and CCL5-dependent.

5, Similarly, they should demonstrate that in the BMDM-fibroblast co-culture system, CCL5 is responsible for the reduced pro-fibrotic gene expression (Fig. 4G to J), or just correlative.

6, More problematic issues are with Fig. 6:

First, the biochemical analysis of WWP2-IRF7 interaction seems quite weak (Fig. 6a). More mapping studies should be performed to confirm such interaction;

Second, the claim about the mono-, or non-degradative ubiquitin conjugation of IRF7 (both in the abstract and text) is not convincing by the data in Fig. 6C. More experimental data are needed to support such claims.

Third, they should compare the nuclear translocation of IRF7 (both normal and p-IRF7) using the method in Fig. 6G and H.

Fourth, the interpretation of Fig. 6J is problematic. The fast-moving band of p-IRF7 may actually represent phospho- (or other forms of) modification, instead of reduced phosphorylation as the authors claimed.

Fifth, more importantly, the authors should provide conclusive genetic evidence that IRF7 indeed is the direct target and mediator of WWP2 in macrophages.

Sixth, as they showed in Fig. S12, TGF- β indeed promoted more M1 to M2 shift in WWP2-deficient monocytes. They should perform direct comparison between IRF7 and Smad2 as they previously published on the WWP2-Smad2 axis (ref. 37) to mutually exclude one target from another.

POINT BY POINT REBUTTALS

Reviewer #1 (Remarks to the Author)

Chen *et al.* present a work on the role of E3 ligase WWP2 in myeloid cells in mouse model of Ang II cardiomyopathy. This is a follow up study of previously published data on WWP2^{mut} mice in the same model. In the current study, authors analyze in details phenotype of myeloid cells not only in WWP2^{mut} mice but also in LysM-Cre x WWP2-flox (newly generated). Authors find that expression of WWP2 in myeloid cells control accumulation of profibrotic macrophages and Ly6c(hi) monocytes in the heart and thus cardiac fibrosis and heart failure in Ang II model. On molecular level WWP2 regulates IRF7 pathway and monocyte polarization. Great part of the data is based on scRNAseq analysis.

The concept of macrophage involvement in heart failure is well known and this work provides a novel molecular insight. In general, molecular part was performed carefully providing high quality data. On the other hand, there are certain open questions mostly related to pathophysiology of the model.

Major concerns:

1. in both transgenic mice, there is reduced influx of monocytes and macrophages at d7. Experiments do not explain this observations but it seems that regardless of molecular signature, the number of cardiac monocytes/macrophages at d7 is responsible for the phenotype at d28. Does WWP2 in myeloid cells affect their migration, proliferation, survival etc?

Answer: We thank the reviewer for finding our manuscript insightful and for his/her suggestions on the pathophysiology related to the model – we have taken into account all the reviewer's recommendations and revised the manuscript accordingly.

According to the reviewer's suggestion, we performed MTT assay to analyse the viability of BMDMs (new **Supplementary Figure 5b-c**). WWP2^{Mut/Mut} BMDMs showed similar cell proliferation compared to WT controls in basal and LPS/IFN γ conditions, suggesting that WWP2 mutation in macrophages does not affect their survival. Regarding cell migration, we performed a scratch wound assay and showed that WWP2 deficiency does not affect the stretch mediated migration in BMDMs (**Following Panel d**). Furthermore, the single cell RNA seq data did not show any significant changes in transcripts belonging to 'cell proliferation' pathway between the experimental conditions (WT control, WT Ang-II infusion, and WWP2^{Mut/Mut} Ang-II infusion) throughout the macrophage subpopulations (**Supplementary Figure 5a**). These additional data support the previous observations and are now reported in the revised manuscript (page 8, line 200-202; see revised **Supplementary Figure 5**).

Supplementary Figure 5. WWP2 and macrophage proliferation and scratch-mediated migration. **a.** Percentage of macrophages in G2/M/S phase in experimental groups (WT control, WT post 7 days Ang-II infusion, and WWP2^{Mut/Mut} post 7 days Ang-II infusion) for clusters 0, 1, 2, 3, 5, 6, and 7. **b-c.** MTT assay showed the relative cell viability in WT and WWP2^{Mut/Mut} BMDMs cultured (**b**) in conditional media and (**c**) after LPS/IFN γ treatment (100ng/ml; 10ng/ml, 4hrs) for different time (hours) (Mann-Whitney U test, n=4-9 for each group).

d. Representative images (*left*) and quantification (*right*) of wound closure scratch assay in a monolayer of WT and WWP2^{Mut/Mut} BMDMs (Mann-Whitney U test, n=12 for each group).

2. Is there a difference in these cells (in the heart and in periphery) in normotensive mice?

Answer: We address the reviewer’s point in two parts. First, we focused on cardiac macrophages and then we measured myeloid cell numbers (monocytes and neutrophils) more generally in periphery. **Figure 2e** and **Figure 3d** show cardiac macrophage numbers and activation markers in normotensive mice according to WWP2 genotype. Compared with WT normotensive mice, the WWP2^{Mut/Mut} normotensive mice showed similar proportion and expression of cardiac macrophage M1/M2-like markers (see also below; **panels a** and **b**). Furthermore, scRNA-seq analysis between normotensive WT and WWP2^{Mut/Mut} cardiac macrophages showed no differentially expressed genes (DEGs) at 5% FDR. These results were further confirmed in the WWP2^{Mac} model whereby we reported a similar proportion of cardiac macrophages between normotensive Ctrl and WWP2^{Mac}. (**Figure 4g** and **panel c**, below). In the revised manuscript we add Supplementary Figure 16f (**Panel d**, below), which shows no difference in M1/M2 markers between normotensive WT and WWP2^{Mac} cardiac macrophages. As per the WWP2^{Mut/Mut} model, WWP2-deficient macrophages did not show DEGs at 5% FDR, which overall confirm an unchanged macrophage infiltration and activation in the normotensive mice upon genetic targeting of WWP2. For clarity, we summarise data in normotensive below (**panels a-d**) and include an additional panel (**Supplementary Figure 16f**).

Figure legend. The relative macrophage number and M1/M2-like gene expression in sorted cardiac macrophages from normotensive hearts. a. Relative percentage of cardiac macrophages (live, CD45+CD64+CD11b+F4/80+Ly6G-) in normotensive WWP2^{Mut/Mut} hearts with respect to WT. P-value calculated by non-parametric Mann-Whitney U test, n=8-9 for each group, dot plots with means ± SD. **b.** Quantitative RT-PCR measuring mRNA expression of selected pro-inflammatory and homeostatic/repair genes in macrophages sorted from WT and WWP2^{Mut/Mut} mice heart without Ang-II infusion (normotensive). Data are represented as dot plots with means ± SD. n=5-8 per experimental group. P-values were calculated by non-parametric Mann-Whitney U test. **c-d.** Relative percentage of cardiac macrophages (**c**) and mRNA expression of M1/M2-like genes (**d**) between Ctrl and WWP2^{Mac} normotensive mice (also listed at **Supplementary Figure 16f**).

Systemically, we evaluated the absolute number of neutrophils and monocytes in blood from both WT and WWP2^{Mut/Mut} mice. In normotensive mice, WWP2^{Mut/Mut} showed similar neutrophil and monocyte numbers when compared with WT mice. We report these data below and add them as part of the revised manuscript (new **Supplementary Figure 4e-f**, page 8, line 194-197). The absolute number of cells was evaluated by flow cytometry and calculated using the precision count beads (BioLegend, 424902), with the formula included in the revised **methods**.

d Blood

Supplementary Figure 4d. Representative flow cytometry plots of the gating used to characterize neutrophils (Live, CD45+ Ly6G+ and live, CD45+ CD11b+ Ly6C+) in blood. **e-f.** Quantification of blood neutrophils (**e**) and monocytes (**f**) number (/ml) in WT and WWP2^{Mut/Mut} mice with or without Ang II infusion (n=4-5 for each group, Mann-Whitney U test, dot plots with means ± SD).

3. What about other cardiac infiltrates (T, B cells, neutrophils etc.) in WWP2^{Mac} mice?

Answer: According to the reviewer's suggestion, we have measured the absolute number of cardiac neutrophils, monocytes/macrophages, T cells and B cells from Ctrl (controls) and WWP2^{Mac} mice (new **Supplementary Figure 16e**, and reported below). The number of monocyte/macrophages was increased with Ang II-infusion at 7 days but reduced in WWP2^{Mac} heart with respect to Ctrl mice (P=0.032). There were no significant differences in cardiac neutrophils, T cells and B cells, between Ctrl and WWP2^{Mac} mice after 7 day-infusion of Ang II. This suggests that the immune cell regulation by WWP2^{Mac}, following Ang II treatment, is mostly affecting cardiac macrophages.

Supplementary Figure 16e. Quantification of cardiac neutrophils, monocyte/macrophages, T cells and B cells number (/heart) in Ctrl and WWP2^{Mac} mice with or without Ang II infusion for 7 days (Mann-Whitney U test, n=4-5 for each group).

4. *the study is performed on a model of cardiomyopathy but no data on cardiomyopathy and cardiomyocyte hypertrophy is given.*

Answer: We have previously shown that Ang II-infusion increased LVMI in control mice, which was reduced in WWP2^{Mut/Mut} [Figure 3g in Chen *et al.* [1]]. According to the reviewer's suggestion, we have measured LVMI in the WWP2^{Mac} mice. We report that, despite a trend towards lower LVMI in WWP2^{Mac} mice, our monocyte/macrophage WWP2 knock-out model does not show a significant reduction of LVMI (new **Supplementary Figure 16a**, and below). We also evaluated cardiomyocyte cell size by wheat germ agglutinin (WGA) staining. As expected, the mean myocyte cross-sectional area (CSA) was increased after Ang II treatment, which was significantly reduced in cardiomyocytes isolated from WWP2^{Mac} hearts (new **Supplementary Figure 16b**, below).

Supplementary Figure 16. a. Left ventricular mass index (LVMI, mg/g) was compared between Ctrl and WWP2^{Mac} mice before and after Ang II-infusion for 7 days. **b.** WGA staining (red) of Ang II infused hearts show hypertrophic myocytes (*left*) with relatively higher mean cell volume, a histological feature significantly reduced in WWP2^{Mac} mice (*right*, n=4-6, Mann-Whitney U test, means ± SD). Scale bar: 50 µm.

5. *WWP2 mRNA and protein levels should be shown for cardiac macrophages sorted from WWP2Mac mice at d7 (see Fig. 4)*

Answer: According to the reviewer's suggestion, we have measured WWP2 mRNA and protein levels in cardiac macrophages sorted from WWP2^{Mac} mice hearts at day 7 (revised **Supplementary Figure 15b**, and below). Using Western Blotting, we also report the lack of WWP2 FL and WWP2 N bands in macrophages sorted from Ang II-induced hearts (**Supplementary Figure 15b, left**, and below). The mRNA expression of floxed WWP2 was also significantly reduced in macrophages sorted from hearts with Ang II-infusion for 7 days (**Supplementary Figure 15b, right**, and below). As per cultured BMDMs and spleen derived macrophages (**Supplementary Figure 15c-d**), the collective data support the deletion of WWP2 from macrophages of WWP2^{Mac} mice.

Supplementary Figure 15. Generation of WWP2^{flx/flx}LysM^{cre} mice. **a.** Schematic representation of WWP2 gene targeting to obtain mice with simultaneous insertion of loxP sites bracketing exon 3. **b.** Representative Western Blot (*left*) for WWP2 in sorted cardiac macrophages isolated from control and WWP2^{Mac} mice. Relative WWP2 mRNA levels (*right*) in sorted cardiac macrophages from WWP2^{Mac} and Ctrl mice measured by quantitative RT-PCR; n=3 per experimental group. Mann-Whitney U test means ± SD. **c-d.** Representative Western Blot of WWP in Ctrl and WWP2^{Mac} mice BMDMs (**c**) and spleen-derived macrophages (**d**), respectively.

Reviewer #2 (Remarks to the Author)

In the manuscript by Chen, *et al.*, the authors address the role of WWP2, an E3 ubiquitin ligase, in the progression of cardiac fibrosis. The authors utilize whole body WWP2-mutant and macrophage (*LysM^{cre}*) conditional WWP2-flox deletion models to address the role of WWP2 in monocyte/macrophage lineage. They use scRNA-seq to describe changes in myeloid cell gene expression following ANGII-mediated cardiac inflammation, including gene expression programs associating CCL5 and IRF7 pathways to loss of WWP2 in monocyte and macrophage populations. A variety of biochemical assays were performed both *ex vivo* and *in vitro*, the authors show a mechanism by which WWP2 binds directly to IRF7 in myeloid cells to regulate its mono-ubiquitination. In the absence of WWP2, there is reduced IRF7 present in the cell, which is needed for the expression of CCL5 through direct chromatin interaction. This pathway leads to attenuation of CCR2⁺ classical monocyte recruitment and reduction in the formation of fibrosis in knockout mice.

Overall this manuscript uncovers a novel connection between WWP2 and its effect on myeloid cells. Mechanistic assays are a major strength of the paper and convincingly link the proposed connection between WWP2-IRF7-CCL5. However, there remains a few lingering questions that would help strengthen the *in vivo* mechanism and the relationship of WWP2 in immune and non-immune cells.

Major:

1) What is the steady state (or non-inflamed) effect of *LysM^{cre} WWP2^{flox}* on cardiac macrophages and circulating monocytes and neutrophils? It would be helpful to know whether local or systemic defects are present prior to the ANGII infusion experiments.

Answer: We thank the reviewer for finding our mechanistic studies compelling. We have considered the Reviewer's constructive suggestions and revised the manuscript accordingly.

The first major point was also partly emphasized by Reviewer #1. To address this, we have measured the proportion of cardiac macrophages and representative cardiac macrophage M1/M2-like markers in normotensive mice. Compared with WT normotensive mice, the WWP2^{Mac} normotensive mice showed similar percentage and expression of M1/M2-like markers. These data have been now included in the revised manuscript as **panel a (Figure 4g)** and new **Supplementary Figure 16f** (see also below).

a. Quantification analysis shows relative percentage of cardiac macrophage (live, CD45+CD64+CD11b+F4/80+Ly6G-) in WWP2^{Mac} hearts with respect to Ctrl normotension hearts. P-value calculated by non-parametric Mann-Whitney U test, n=8-9 for each group, dot plots with means \pm SD.

Supplementary Figure 16f. Quantitative RT-PCR measuring mRNA expression of selected pro-inflammatory and homeostatic/repairatory genes in macrophages sorted from Ctrl and WWP2^{Mac} mice heart without Ang-II infusion for 7 days. Data are represented as dot plots with means \pm SD. n=5-8 per experimental group. non-parametric Mann-Whitney U test, dot plots with means \pm SD.

In addition, we evaluated the number of neutrophils and monocytes in blood from both WT and WWP2^{Mac} mice. Normotensive WWP2^{Mac} mice showed similar neutrophil and monocyte numbers with WT mice in blood (**Supplementary Figure 16c**, and reported here on the right).

Supplementary Figure 16c. Quantification of circulating neutrophil and monocyte numbers (/ml) in WT and WWP2^{Mac} mice without Ang II infusion (P-value by Mann-Whitney U test, n=3 in each group, dot plots with means \pm SD).

2) Please report macrophage data as total cells/heart and percentage of CD45⁺, and blood cells should be presented as cells/mL blood. In keeping with this data reporting approach, could the authors also provide immune cell counts for the hearts in the ANGII experiments that are presented? The flow plots presented in Figure 2 are difficult to interpret because even non-macrophages appear to be dramatically reduced (such as CD11b⁺ F4/80⁻), and the negative populations appear to be very different autofluorescence. This makes the data presented as a % of total cells difficult to fully interpret. From experience, knowing total macrophage counts/heart and percentage of CD45⁺ helps to draw conclusions regarding expansion or lack- thereof in tissues.

Answer: We thank the reviewer for this useful suggestion regarding data reporting. According to the Reviewer's recommendation, we measured the macrophage counts/heart and the percentage of CD45⁺ in both WT and WWP2^{Mut/Mut} mice. Using beads, we calculated the absolute cardiac macrophage numbers and confirmed that WWP2 deficiency suppressed the Ang II-induced increase in cardiac macrophages observed in WT hearts (new **Supplementary Figure 4g**, and below). Accordingly, both the percentage of CD45⁺ cells in live single cells from whole heart, and the percentage of macrophages in CD45⁺ cardiac cells were decreased in WWP2^{Mut/Mut} mice after 7 day-infusion of Ang II; see new **Supplementary Figure 4h**, and below. We agree with the Reviewer that these new data help to draw definitive conclusions on the effect of WWP2 on cardiac macrophage infiltration.

Supplementary Figure 4. g. Flow cytometry with beads (*left*, square), which was used to measure the absolute macrophage numbers in WT and WWP2^{Mut/Mut} mice with or without Ang II infusion for 7 days (*right*). **h.** Quantification shows reduced percentage of cardiac CD45⁺ cells (*left*) and percentage of macrophages (*right*) in CD45⁺ cells (live, CD45⁺CD64⁺CD11b⁺F4/80⁺Ly6G⁻) in WWP2^{Mut/Mut} hearts with respect to WT. n=6 for each group.

3) Finally, did the investigators perform any *i.v.* labeling controls to determine whether all subsets of immune cells are indeed infiltrating into the heart or merely accumulating along the vasculature?

Answer: The reviewer highlights a pertinent technical point about discriminating between peripheral immune cell dynamics in the vasculature vs cardiac infiltrates. We did not use antibodies to label circulating immune cells through macro- and microvasculature. All experiments related to our cardiac perfusion procedure were carried out by a single (expert) operator. Although occasional faulty perfusion may increase the amounts of the intravascular immune cell contaminants, we expect this to be negligible for tissue macrophages (the primary target immune cell of WWP2), which are rarely found in blood [2].

4) The authors describe defects in monocyte recruitment in the whole body WWP2^{-/-} and conditional knockout systems but do not perform monocyte recruitment assays. Alternative possibilities to defective recruitment could be that WWP2 monocytes are dying within the tissue, or potentially re-circulating following entry. Could authors provide supporting data for their conclusion?

Answer: We agree with the Reviewer that we cannot conclusively rule out additional mechanisms (cell death, re-circulating following entry). As suggested, we explored alternative pathways underlying macrophage reduction in WWP2^{Mut/Mut} hearts by performing additional experiments and interrogating further some of the existing data (below): (i) cultured WWP2^{-/-} BMDMs showed similar proliferation and migration with WT cells (this was added as a new **Supplementary Figure 5b-c**, and reported below), (ii) compared with WT controls, WWP2^{Mut/Mut} mice had similar number of the monocytes in peripheral blood (this was added as a new **Supplementary Figure 4e-f**, and reported below); (iii) single cell RNA-seq analysis did not show any significant difference in cell proliferation (% of G2/M/S) in any of the cardiac macrophage subpopulations between WT and WWP2^{Mut/Mut} mice (**Supplementary Figure 5a**, and reported below), (iv) absence of pathways related to any form of cell death in cardiac macrophage

scRNA-seq data (Figure 3b and Supplementary Figure 9a). Collectively, these data suggest that WWP2 has no evident effect on monocyte cell death.

We discuss this point in our revised manuscript and emphasize that WWP2 reduces the expression of CCL5, a cytokine that plays a key role in monocyte/macrophage recruitment in cardiac injury, which is in line with the experimental and *in silico* results obtained above (Page 18, line 527-529).

Supplementary Figure 5. b-c. MTT assay showed the relative cell viability in WT and WWP2^{Mut/Mut} BMDMs cultured (b) in conditional media and (c) after LPS/IFN γ treatment (100ng/ml; 10ng/ml, 4hrs) for different time (hours) (Mann-Whitney U test, n=4-9 for each group).

Supplementary Figure 4. e-f. Peripheral neutrophil (e) and monocyte (f) numbers (/ml) in WT and WWP2^{Mut/Mut} mice with or without Ang II infusion (Mann-Whitney U test, n=4-5 for each group).

Supplementary Figure 5. a. Percentages of macrophages in G2/M/S phase in experimental groups (WT control, WT post 7 days Ang-II infusion, and WWP2^{Mut/Mut} post 7 days Ang-II infusion) for clusters 0, 1, 2, 3, 5, 6, and 7.

Figure 3b. Grouping of the top downregulated pathways identified by gene set enrichment analysis (GSEA) of DEGs in macrophages isolated from WWP2^{Mut/Mut} and WT mice after treatment with Ang-II infusion (7 days) from our scRNA-seq data (Figure 2a). A significance score for each pathway is calculated as the product of the normalized enrichment score (NES) by GSEA and the ratio of overlapping genes with the gene set.

Supplementary Figure 9a. Upregulated pathways in cardiac macrophages and macrophage subsets associated with WWP2 deletion. a. Top upregulated pathways from gene set enrichment analysis (GSEA) of differentially expressed genes (DEGs) between macrophages isolated from WWP2^{Mut/Mut} and WT mice. Pathways are grouped into “Homeostasis” and “Calcium and acid metabolism” meta-pathways. Scores for each pathway are calculated as the product of the normalized enrichment score (NES) and the ratio of overlapping genes with the entire canonical gene set.

5) Similarly, monocle data predicting monocyte differentiation paths following entry into the heart is presented in a somewhat unconventional fashion. Could the authors present this data on a UMAP projection? Based on the limited data shown, it also appears like cluster 6 and cluster 5 do not appropriately fit in the trajectory well and might be more suitable to be excluded. In addition, it has become increasingly common to provide multiple trajectory analysis approaches for scRNA-seq datasets. Adding slingshot or RNA-velocity analysis might help strengthen arguments regarding the interconnectedness between clusters and predicted differentiation pathways.

Answer: According to the Reviewer's suggestion, we performed pseudotime analysis for 5 clusters (0-4) on a UMAP projection after excluding cluster 5 and 6 (see revised manuscript new **Figure 1h**, and reported below). This shows a pseudotime connecting the 5 clusters with a branching point in the pro-inflammatory cluster C2. C2 links with C3 (stem cell-like cluster), C4 (monocyte cluster), and C0-C1 (AP1/homeostatic cluster), revealing a trajectory connecting the inflammatory and homeostatic clusters. Likewise, pseudotime analysis using Monocle recapitulated a trajectory connecting homeostatic and monocyte clusters with the following order: C0-C1-C2-C3-C4 (new **Supplementary Figure 3c**, and below).

These analyses confirmed the interconnection between cluster 0 to cluster 4, and suggest that the increase in the number of cardiac macrophages following 7 days Ang II infusion is most likely due to Ly6C^{high} monocyte (cluster 4) infiltration. Notably, both Monocle (**Figure 1h**) and Slingshot (**Supplementary Figure 3c**) independently identified the connection between inflammatory (Cluster 2) and homeostatic macrophages (Cluster 0), through the intermediate cluster C1 which represents the transition between the two macrophage subtypes along the monocyte differentiation path. We thank the Reviewer for this suggestion, which indeed strengthened the predicted differentiation pathways in the model.

Figure 1h. UMAP of cardiac macrophages describing developmental trajectories superimposed (*upper*) in clusters from C0 to C4 (*down*). The main trajectories are generated by Slingshot. Colors are determined by the metacell's pseudotime, which is calculated as the average pseudotime over all cells in the metacell.

Supplementary Figure 3c. UMAP of cardiac macrophages with single-cell trajectories from C0 to C3 (*left*, red line) superimposed to single-cell clusters (*right*). The main curves are generated by Monocle analysis and colors are determined by the average pseudotime over all cells.

6) Since there is reduced monocyte recruitment, do tissue resident cardiac macrophages proliferate at a faster rate in the WWP2^{-/-} mice, compared to controls in the steady state or following ANGII?

Answer: This is indeed a plausible scenario that we took into consideration in the revised manuscript. We first interrogated the single cell RNA seq data which showed no significant changes in cell proliferation between the different experimental conditions (i.e., WT control, WT Ang-II infusion, and WWP2^{Mut/Mut} Ang-II infusion) throughout all macrophage subpopulations (Supplementary Figure 5a). In order to experimentally confirm the latter, we then tested cell proliferation in WT and WWP2^{Mut/Mut} BMDMs. This revealed no significant changes (new **Supplementary Figure 5b-c**). To further link these results to the heart, we performed immunofluorescence of CD68 and Ki67 in cardiac sections (new **Supplementary Figure 5d**, and shown below). As expected, WWP2^{Mut/Mut} showed less CD68+ cells in cardiac sections compared with WT mice. However, the CD68+Ki67+ cells did not show a significant difference (P=0.132) between WWP2^{Mut/Mut} and WT hearts after Ang II-infusion. These new data are now included in the revised manuscript (page 8, line 200-207) and in a new **Supplementary Figure 5**.

Supplementary Figure 5d. Representative photomicrographs of immunohistochemical staining of CD68 and Ki-67 in myocardial heart sections from mice after Ang II-infusion (*left*). Arrows indicate CD68+Ki67+ cells. Quantification of CD68+ cells and CD68+Ki67+ cells (*right*). n=3-6 for each group; P-values were calculated by Mann-Whitney U test; data are reported as mean ± SD. Size bar indicates 50 µm.

7) The authors show that conditional depletion of WWP2 in macrophages regulates the fibrotic response to ANGII infusion. The magnitude of protection is equal to the total protection previously found in the total body knockout (Chen, *Nature Communication* 2019). Are the authors able to weigh the differential roles of WWP2 in cardiac fibroblasts against macrophages? Experimentally, it would be valuable to know if replacement of bone marrow in WWP2-mutant mice with WT donor cells (thus creating WT-macrophages but KO stroma) is sufficient for protection (or loss) from the fibrotic response. In a similar line of experiments, performing mixed bone marrow chimera experiments (50% WT, 50% KO) comparing cell intrinsic versus extrinsic influences of WWP2-deletion on monocytes and their ability to be recruited into the heart may prove to be valuable to the interpretation of this project.

Answer: The Reviewer highlights a relevant point which requires putting into perspective the studies we have been carrying out on WWP2 and fibrosis for the last 7 years. In our previous study [1], we have demonstrated that the regulation of WWP2 in cardiac fibroblasts contributes to the cardiac fibrosis. In the present study, we report the regulation of WWP2 in cardiac macrophages, and show that such WWP2 regulation is key to cardiac fibrosis using conditional knockout mice.

These findings evidently bring the question of relative contribution of WWP2 in fibroblasts vs macrophages through which WWP2 exerts its regulatory role on cardiac fibrosis. While we currently actively work on this question in different tissues and model organisms, the results presented in this study (i.e., WWP2 regulating Ccl5-expressing Ly6C^{high} monocyte infiltration and cardiac fibrosis) will not be invalidated by determining the relative regulatory contribution of WWP2 in immune cells vs stroma. In fact, our unpublished work suggests that the immune vs stroma contribution of WWP2 to tissue fibrosis is organ-specific. We agree with the Reviewer that the suggested bone marrow chimera experiments may prove to be valuable to the interpretation of our current studies on WWP2. However, the suggested experiments will not add mechanistic insights onto the macrophage-mediated effect of WWP2 on cardiac fibrosis (the current study).

Minor:

1. Page 14, Line 419: I believe Figure 5J is incorrectly referenced when the authors are referencing Fig 6D.

Thanks for spotting this mistake, which has been amended.

2. Method description for monocle appears to have description errors in terms of excluded data clusters and start point for pseudotime.

Thanks for highlighting this. The method description should read as “Outlier clusters S1 and S2 were removed prior to calculation of pseudotime. Ordering of cells was performed using the UMAP dimension reduction and cluster C5 was chosen as the root cells” (method, Page 5, line 218-220).

3. How were cardiac macrophages sorted for qPCR analysis? Were monocyte excluded?

In our study, the cardiac monocyte compartment (including monocyte and macrophages) is identified by CD45+CD64+CD11b+ cells [3]. We analysed the gene mRNA levels in macrophages sorted from digested hearts and the marker for cardiac macrophages which we used to sort cells was CD45+CD64+CD11b+F4/80+Ly6G- [4, 5]. Lavine *et al.* [6] showed that the cardiac monocytes present low F4/80 expression, and cardiac macrophages have high F4/80 expression. In the revised manuscript, we now clearly indicate the gating strategy used to sort cardiac macrophages that were used for the qPCR analysis (Page 10, line 289 and method Page 4, line 176).

4. *Figure 3J – please describe how fluorescence images were normalized and intensity was calculated.*

Image J was used to analyse the fluorescent image with ACTA and Col1 staining [7]. After choosing a field as a region of interest (ROI), the fluorescence intensity (MFI) was quantified, while a non-fluorescent area of the same images was measured, and used as the MFI background. Cell counting from the same field was automatically assessed by DAPI staining. The final intensity of positive cells per cell is calculated as $(MFI_{ROI} - MFI_{background}) / \text{total cell number in field}$. We now clearly indicate this in the revised manuscript (Page 3, line 1110-1117).

5. *A recent and potentially relevant citation for immune mechanisms of cardiac fibrosis should be included, Revelo *et al.* Circulation Research 2021*

We agree with the Reviewer, and have now added this useful reference in the revision.

6. *Page 35 line 17-19: Statement needs revised from how it currently reads; I believe the authors are referencing the breeder pairs only.*

We clarify that WWP2^{flox/flox} males were crossed with WWP2^{flox/flox} LysM^{WT/Cre} females to generate WWP2^{flox/flox} LysM^{WT/Cre} mice (WWP2Mac) and Cre-negative WWP2^{flox/flox} littermates for controls (Ctrl). (This breeding scheme prevents embryonic deletion of WWP2 by Cre-expressing females). We have now amended this in the revised manuscript (Method, Page 1, line 17-19).

Reviewer #3 (Remarks to the Author)

In this paper, Chen et al described a role of the E3 ligase WWP2 in heart fibrosis via regulating a Ly6C-high subpopulation of monocytes. By using single cell profiling, they found that Ly6C-high monocytes were increased upon angiotensin II treatment, which were reversed in WWP2-abalated mice. WWP2-deficient monocytes showed reduced expression of several chemokines like CCL5, and other pro-inflammatory cytokines such as IL6, TNFa, and IL1b, but increased expression of IL4, which was further confirmed by conditional deletion of WWP2 in monocytes. Transcriptional profiling suggested that IRF7 served as a target which was decreased in WWP2-deficient monocytes, and could drive the expression of itself (IRF7) and CCL5. They further provided results showing that WWP2 interacted with IRF7, promoted its ubiquitination and nuclear translocation. The results seem to agree with their previous publication showing that WWP2 controls heart fibrosis via regulating the ubiquitination of Smad2 in cardiac fibroblasts (ref. 37, NC 2019). However, it is concerned that the key data are preliminary and largely correlative, and therefore not entirely convincing at the current stage.

Reply to the general comment:

We thank the Reviewer for the time spent assessing our manuscript and for his/her feedback and suggestions.

With due respect, we disagree with the general statement that our “*key data are preliminary and largely correlative*”, and we argue which “*key data*” this Reviewer is referring to. The other 2 Reviewers (see detailed comments above, and in short below) have confidence that we provided the necessary experimental evidence demonstrating that main finding, i.e., that WWP2 regulates a Ly6C^{high} subpopulation of monocytes in heart fibrosis and that WWP2 is a regulator of IRF7-mediated Ccl5/Ly6c^{high} monocyte axis. While providing several suggestions and criticisms, both Reviewers attested to the high quality and mechanistic nature of our data, and commented (*verbatim*): “*In general, molecular part was performed carefully providing high quality data*” (R#1) and “*Mechanistic assays are a major strength of the paper and convincingly link the proposed connection between WWP2-IRF7-CCL5*” (R#2).

That said - we also wish to highlight that the most of the Reviewer’s crucial criticisms are focused to very specific points related to protein ubiquitination and/or cytokine function. As such, these narrow criticisms do not pay justice to the large bulk of *in vivo*, *ex vivo*, *in vitro* and single-cell data presented in support of the main finding of our study. While we acknowledge the importance of protein ubiquitination and/or cytokine function here, we believe that some of the suggested experiments by this Reviewer are redundant, unessential, and do not challenge our central hypothesis. For instance, establishing the role of previously well-characterized IL-6, TNFa, and IL-1 cytokines in our system, does not disprove (nor validate/strengthen) our main discovery.

Similarly, the role of CCL5 in monocyte/macrophage recruitment has been extensively established. Our study is not meant to re-discover these findings. Instead, our study aimed to offer a novel perspective for the regulatory role of WWP2 in monocyte/macrophage-mediated cardiac fibrosis in NICM, and we believe this will open the door for future and more extensive experiments (by us and other groups) to detail all proteins (e.g., cytokines, other chemoattractants) and signaling regulated by WWP2 in cardiac macrophages in fibrosis.

Below, we provide detailed rebuttals and data to address each point raised by the Reviewer. In addition, according to the Reviewer’s suggestions, we also tuned down some of the statements (including those in the Abstract) that can be perceived as strong, in an attempt to avoid misleading or overreaching conclusions in our manuscript.

Major points:

1, The observed reduction of Ly6C-high monocytes in WWP2-deficient mice (Fig. 1) suggested that WWP2 may function in an early phase of inflammation via a direct effect on the production of pro-inflammatory cytokines such as IL-6, TNFa, and IL-1 (Fig. 3 and Fig. S10 to 11), rather than an indirect effect on CCL5-mediated infiltration. The authors should carefully examine the roles of these cytokines in their model.

Answer: We agree with the Reviewer on the important function of these (and other) cytokines in cardiac fibrosis, and we cannot exclude their contribution to the early phase of inflammation, which is well documented. IL-6 is showed to be important in the regulation of collagen by cardiac fibroblasts [8]. TNFa signalling acts as a significant determinant of the downstream repair process to mediate collagen composition and alignment in the scar following MI [9]. IL-1 β and TNF- α may contribute to ventricular dilation and myocardial failure by promoting the remodelling of interstitial collagen [10]. Indeed, cardiac macrophages are considered the main source of these cytokines, which play a critical role in the pathogenesis of cardiac fibrosis. Hence, the role of these cytokines in cardiac fibrosis has

been previously established and it would be redundant to duplicate previously published experiments. To “examine the roles of these cytokines in [our] model” is and was not the central hypothesis of our study.

Besides, we have already shown that WWP2 regulates these cytokines (**Figure 3** and **supplementary Figure 11-13**). We took the Reviewer point into account, and, in our revised manuscript, we provide a more detailed discussion, and additional references detailing the known roles of these cytokines in cardiac fibrosis (Page 19, line 546-550).

2, Similarly, the authors claimed that CCL5 was a critical mediator for their observed phenotypes (Figs. 2 and 4), but they did not provide direct evidence to support such claims in both in vitro and in vivo systems.

Answer: We demonstrated that WWP2 deficiency results in decreased CCL5 expression in cardiac macrophages, especially in Ly6C^{high} monocytes, and in reduced secretion of CCL5 by BMDMs. The “*observed phenotypes*” mediated by Ccl5 for which the Reviewer states we “*did not provide direct evidence*” are - monocyte recruitment in fibrotic heart – (shown in **Figure 2b-e**).

The chemotactic role of CCL5 has been extensively demonstrated in different murine models and reported previously. Ccl5 is critical for macrophage recruitment and infiltration to tissues [11-14], and the Ccl5-mediated macrophage recruitment associates with myocardial infarction [15] and cardiac dysfunction [16]. Some of the key references on the known CCL5 functions were already cited and discussed in our manuscript. Therefore, our data supporting Ccl5-mediated recruitment of Ly6C^{high} monocytes downstream WWP2 are novel in context of WWP2-regulated cardiac fibrosis. The previously and firmly established role of CCL5 in monocyte recruitment will be redundant to be duplicated here.

3, It is generally believed that the M1 to M2 transition of macrophages, as they observed in WWP2-deficient mice and in vitro culture system (Fig. 3, and Fig. S9 to 12), will promote late-stage fibrosis/repair response. This observation seems to contradict with the authors' claims.

Answer: Some reports indeed suggested that M2-like macrophages promote late-stage fibrosis/repair response [17]. However, there are emerging controversies over this theory [18], which are in part due to the use of a too simplistic and binary M1/M2 classification, which historically has been derived from *in vitro* studies. More importantly, in recent years, this naïve classification was revisited by single cell transcriptomics analysis of cardiac myeloid cells, and our findings (i) are in line with the studies on cardiac single cell RNA-seq, in particular those coming from the Epelman group [3, 19], (ii) add mechanistic insights into the pro-fibrotic role of Ly6C^{high} early infiltrating monocytes and (iii) reveal a master regulator (WWP2) of the recently established cardiac macrophage polarization during NICM. In summary, resident/homeostatic/repair macrophages express high Timd4 and low Ccr2 and seem to be negatively correlated with fibrosis, while monocyte-derived Ly6C^{high}/pro-inflammatory cluster express high Ccr2 and low Timd4 and positively correlate with fibrosis. Crucially, we found a regulatory effect of WWP2 on Ly6C^{high} monocyte subgroup enriched in ECM backbone genes. Hence our results validate the current understanding of the state of activation of cardiac macrophages and reveals a novel role of WWP2 through Ccl5/IRF7 axis (see also discussion, second and fourth paragraph for detailed description of our results in the context of cardiac macrophage activation/polarization and cardiac fibrosis).

4, Another concern is from the authors' observation that WWP2 deficiency significantly reduced the numbers of Ly6C-high monocytes (Fig. 2C to E and 4G to I), which should be experimentally tested to see whether such cell number reduction is IRF7- and CCL5-dependent. Similarly, they should demonstrate that in the BMDM-fibroblast co-culture system, CCL5 is responsible for the reduced pro-fibrotic gene expression (Fig. 4G to J), or just correlative.

Answer: With due respect, we believe this is another satellite question that is not essential to prove/disprove our hypothesis. Our study shows the regulation of IRF7/CCL5 by WWP2 at early (inflammatory) stage of cardiac fibrosis. Whether “cell number reduction is IRF7- and CCL5-dependent” or “CCL5 is responsible for the reduced pro-fibrotic gene expression” are questions that are not related to WWP2 but more to CCL5 and IRF7 functions *per se*. Furthermore, the Reviewers' interrogations about CCL5 and IRF7 have been, to an extent, already published (see rebuttal to point 2 for CCL5). Similarly, IRF7 deficiency yields reduced expression of pro-inflammatory cytokines (IL6, TNF α and I β) in tissue and macrophages [20, 21]; IRF7 knock-out mice attenuates tissue fibrosis *in vivo* [22].

In the revised manuscript, we have extended the Discussion and included these published data on IRF7 function in macrophages and tissue fibrosis, which we believe are in line with the primary findings of our manuscript, suggesting that the IRF7/CCL5 pathway might be the potential mechanism contributing to macrophage infiltration and fibrotic phenotype downstream of WWP2.

More problematic issues are with Fig. 6:

5. First, the biochemical analysis of WWP2-IRF7 interaction seems quite weak (Fig. 6a). More mapping studies should be performed to confirm such interaction;

Answer: The data shown in **Fig. 6a** show a clear band in the WB for the WWP2-IRF7 interaction. Here we highlight that this assay has been carried out in primary cultured macrophages, which yields a readout for endogenous WWP2-IRF7 interaction. This technique does not over-express the target protein and may result in “weaker” bands due to endogenous expression of target proteins. We deliberately chose this approach as when overexpressed, proteins may artificially regulation with one another in ways that do not occur at endogenous expression levels [23].

According to the reviewer’s suggestion, to strengthen our data, we have carried out an additional experiment to further confirm WWP2-IRF7 interaction, and exogenously expressed WWP2 isoforms in HEK293T cells, confirming IRF7 co-immunoprecipitation with Flag-tagged WWP2-N isoform, and with Flag-tagged WWP2-C isoform (weaker) (new **Supplementary Figure 18c**, and below).

Supplementary Figure 18. c. HeK293 cells were transfected with WWP2-Flag isoforms. Co-immunoprecipitation shows a direct interaction of IRF7 with WWP2, especially with N-isoform. Lysates were subjected to immunoprecipitation with anti-FLAG antibodies, followed by western blotting probed with antibodies as indicated.

6. Second, the claim about the mono-, or non-degradative ubiquitin conjugation of IRF7 (both in the abstract and text) is not convincing by the data in Fig. 6C. More experimental data are needed to support such claims.

Answer: In addition of the data shown in **Figure 6C**, we have now combined other data and considered the results of several experiments to support the claim of non-degradative ubiquitin conjugation of IRF7, as follows.

First - we identified (and hence proposed) that IRF7 is a new substrate of WWP2 based on scRNA-seq analysis in primary cardiac tissue. We then showed endogenous physiological interaction between IRF7 and WWP2 in primary cultured macrophages (**Figure 6a**). *Second* – this is confirmed using exogenously expressed WWP2-FLAG (new **Supplementary Figure 18c**). *Third* - we observed no increased IRF7 protein expression in WWP2^{-/-} macrophages compared with WT cells (**Figure 5i**). Therefore, taken together these data are supportive of WWP2 modulating IRF7 function via a non-degradative mechanism in macrophages.

Based on the supportive evidence described above, we then show that the bands of mono-ubiquitin and multi-ubiquitin conjugation of IRF7 were relatively reduced in WWP2^{-/-} BMDMs compared with WT cells (**Figure 6c**), since mono-ubiquitination and/or multi-ubiquitination are the main non-degradative modification of E3 ligases on target proteins. Hence, the data in **Figure 6c** should be considered in context of other data shown in our manuscript.

On a related note, and as highlighted above, we want to comment that in our study we carried out experiments to inform the physiological interaction and ubiquitin conjugation of IRF7. We did not use an exogenous expression system to investigate the ubiquitin regulation at IRF7, as the overexpression of E3 ligases might often lead to non-physiological degradation of the substrate (e.g., of IRF7 here). We have detailed this important technical detail in our previous study [1], where we compared/discussed our data with those presented by Soond *et al.* [24] who reported the selective degradation of SMADs (upon TGFβ activation) by specific and exogenously expressed WWP2 isoforms. Using multiple line of evidence from *ex vivo* tissue, *in vitro* (gain of function and loss of function) and *in vivo* murine studies, we unambiguously demonstrated the “physiological” activity of WWP2. This “physiological” activity of WWP2 was indeed consistent with the finding from another *in vivo* study [25], and, notably, this was opposite to the one that was described by Soond *et al.* [24] who relied only on exogenously expressed WWP2 isoforms. Hence, we choose to do not use an exogenous expression system to investigate the ubiquitin regulation at IRF7 by WWP2.

7. Third, they should compare the nuclear translocation of IRF7 (both normal and p-IRF7) using the method in Fig. 6G and H.

Answer: We have now made the suggested experiments and report the new data in the revised **Figure 6g** and **Figure 6i**. The new data show that percentage of nuclear p-IRF7 in WT BMDMs was increased by LPS stimulation, and was significantly lower in WWP2^{-/-} compared with WT BMDMs (**Figure 6i**, and below). These data have been added to the results of the revised manuscript (page 15, line 454-463).

Supplementary Figure 6g. Representative cellular images of in-cell IRF7 and p-IRF7 from 10,000 acquired events by imaging flow cytometry, which show typical externalized and internalized patterns with colocalized or separated distributed IRF7 and p-IRF7. **h-i.** Gating scheme shows IRF7 (**h**) and p-IRF7 (**i**) nuclear translocation in both WT and WWP2^{-/-} BMDMs with LPS treatment (100ng/ml, 4hrs) with a similarity score ≥ 1.5 (*left*). Quantification analysis showed a reduced percentage of nuclear IRF7 in WWP2^{-/-} BMDMs with respect to WT cells upon LPS treatment (*right*). n=4 for each group.

8. For the interpretation of Fig. 6J is problematic. The fast-moving band of p-IRF7 may actually represent phospho- (or other forms of) modification, instead of reduced phosphorylation as the authors claimed.

Answer: We thank the Reviewer for rising this relevant point, and we acknowledge the presence of fast-moving IRF7 and p-IRF7 bands in nuclear fraction of WWP2^{-/-} BMDMs with respect to the p-IRF7/IRF7 bands detected in WT BMDMs. We utilised a widely-used phospho-specific antibody targeting residues surrounding Ser437/438 of IRF7, which is the main phosphorylation modification in WT BMDMs [26]. This band detected in WT at ~55-60kD

position is shifted in WWP2^{-/-} BMDMs, showing a relatively weaker, fast-moving band. We agree that this might be due to other form of protein modifications of IRF7 [27]. However, our data clearly shows that, the effect of WWP2 deficiency results in IRF7 protein modification(s), reduced nuclear IRF7 and p-IRF7, which ultimately lead to dampened transcriptional activity of IRF7 in BMDMs.

We take the Reviewer's point on board, and in the revised manuscript we provide more details on the interpretation of the data shown in **Figure 6j** (Page 15, line 453-454).

Figure 6j. Representative western blot showing IRF7 and p-IRF7 protein distribution in nuclear fractions of WT and WWP2^{-/-} BMDMs with or without LPS (100ng/ml, 4hrs).

9. more importantly, the authors should provide conclusive genetic evidence that IRF7 indeed is the direct target and mediator of WWP2 in macrophages.

Answer: The Reviewer suggests us to provide “conclusive genetic evidence” that that IRF7 is a WWP2 target and mediator in macrophages. Let's recap what we have shown in our manuscript. Our scRNA-seq analysis showed that IRF7 signalling network in cardiac macrophages is regulated by WWP2. We further confirmed the direct regulatory interaction between WWP2 and IRF7, and the modulation of IRF7 activity, mono/heterodimerization, nuclear translocation, ubiquitination by WWP2 by loss of function experiments in BMDMs. We believe that collectively this bulk of data suggests that IRF7 is a direct target of WWP2 in macrophages.

Now, let's move on to what is known on the role of IRF7 in macrophage activation and tissue fibrosis: (1) IRF7 deficiency yields reduced expression of pro-inflammatory cytokines (IL6, TNF α and I β) in tissue and macrophages [20, 21], (2) IRF7 deficiency attenuates tissue fibrosis *in vivo*, and (3) IRF7 represents a link between inflammation and fibrosis in the pathogenesis of fibrotic disease, such as systemic sclerosis [22].

Combining our data on IRF7 regulation by WWP2 with the published data on IRF7 on macrophage function and fibrosis, we propose that “WWP2 regulates IRF7 transcriptional activity in Ccl5 expressing Ly6c^{high} monocytes, which in turns contributes to the pro-inflammatory and pro-fibrotic macrophage activation and infiltration in the fibrotic heart”. Cautiously, we also pointed out that “we cannot exclude a direct effect of WWP2 on homeostasis/repair macrophages through other immunomodulatory transcription factors (c-MAF) and the anti-inflammatory chemokine Ccl24” (page 20, line 580-581). Thus, providing conclusive genetic evidence to link, IRF7 to fibrosis diverts the question from the role of WWP2, which is central to our working hypothesis. Furthermore, the reviewer's suggestion will require additional *in vivo* models that would be seen as out of scope.

10, as they showed in Fig. S12, TGF- β indeed promoted more M1 to M2 shift in WWP2-deficient monocytes. They should perform direct comparison between IRF7 and Smad2 as they previously published on the WWP2-Smad2 axis (ref. 37) to mutually exclude one target from another.

Answer: Indeed, we showed that WWP2 deficiency increased the expression of some genes representative of an M2-like phenotype under TGF β 1 treatment, including *Ii10* and *Folr2* (**Figure Supplementary 12d**). However, based on these data and the results of our scRNA-seq analysis for WWP2-regulated pathways in cardiac macrophages (**Figure 3b**) and the transcription-factor regulon analysis (which didn't show SMADs, **Figure 5a**), we concluded that the regulation of WWP2 in cardiac macrophage is mainly through IRF7 regulon, which might also interact with other signalling pathways including Smad-signaling in fibroblasts, as previously shown [22]. (Page 20, line 582-601). We never stated that the regulation of WWP2 in cardiac macrophage is exclusively through IRF7.

Because we have previously showed that WWP2 regulates Smad2 in fibroblasts [1], the Reviewer now asks us to “mutually exclude one target from another”, i.e., to mutually exclude IRF7 and Smad2 as targets of WWP2. As shown for other E3 ligases [28-30], it is likely that WWP2 have multiple targets in different cell-types, which might interact with each other. While of interest, this specific request is noticeably far-reaching, and we believe that demonstrating whether IRF7 and Smad2 are mutually exclusive targets of WWP2 is totally irrelevant for the main message of our manuscript.

References

1. Chen, H., et al., *WWP2 regulates pathological cardiac fibrosis by modulating SMAD2 signaling*. Nat Commun, 2019. **10**(1): p. 3616.
2. Aronoff, L., S. Epelman, and X. Clemente-Casares, *Isolation and Identification of Extravascular Immune Cells of the Heart*. J Vis Exp, 2018(138).
3. Dick, S.A., et al., *Self-renewing resident cardiac macrophages limit adverse remodeling following myocardial infarction*. Nat Immunol, 2019. **20**(1): p. 29-39.
4. Liao, X., et al., *Distinct roles of resident and nonresident macrophages in nonischemic cardiomyopathy*. Proc Natl Acad Sci U S A, 2018. **115**(20): p. E4661-E4669.
5. Liu, Z., et al., *Analysis of Myeloid Cells in Mouse Tissues with Flow Cytometry*. STAR Protoc, 2020. **1**(1): p. 100029.
6. Lavine, K.J., et al., *Distinct macrophage lineages contribute to disparate patterns of cardiac recovery and remodeling in the neonatal and adult heart*. Proc Natl Acad Sci U S A, 2014. **111**(45): p. 16029-34.
7. Shihan, M.H., et al., *A simple method for quantitating confocal fluorescent images*. Biochem Biophys Rep, 2021. **25**: p. 100916.
8. Melendez, G.C., et al., *Interleukin 6 mediates myocardial fibrosis, concentric hypertrophy, and diastolic dysfunction in rats*. Hypertension, 2010. **56**(2): p. 225-31.
9. Barnette, D.N., et al., *iRhom2-mediated proinflammatory signalling regulates heart repair following myocardial infarction*. JCI Insight, 2018. **3**(3).
10. Siwik, D.A., D.L. Chang, and W.S. Colucci, *Interleukin-1beta and tumor necrosis factor-alpha decrease collagen synthesis and increase matrix metalloproteinase activity in cardiac fibroblasts in vitro*. Circ Res, 2000. **86**(12): p. 1259-65.
11. Vajen, T., et al., *Blocking CCL5-CXCL4 heteromerization preserves heart function after myocardial infarction by attenuating leukocyte recruitment and NETosis*. Sci Rep, 2018. **8**(1): p. 10647.
12. Keophiphath, M., et al., *CCL5 promotes macrophage recruitment and survival in human adipose tissue*. Arterioscler Thromb Vasc Biol, 2010. **30**(1): p. 39-45.
13. Lee, C.M., et al., *C-C Chemokine Ligand-5 is critical for facilitating macrophage infiltration in the early phase of liver ischemia/reperfusion injury*. Sci Rep, 2017. **7**(1): p. 3698.
14. Jongstra-Bilen, J., et al., *Role of myeloid-derived chemokine CCL5/RANTES at an early stage of atherosclerosis*. J Mol Cell Cardiol, 2021. **156**: p. 69-78.
15. Montecucco, F., et al., *CC chemokine CCL5 plays a central role impacting infarct size and post-infarction heart failure in mice*. Eur Heart J, 2012. **33**(15): p. 1964-74.
16. Wang, X., et al., *C-C chemokine receptor 5 signaling contributes to cardiac remodeling and dysfunction under pressure overload*. Mol Med Rep, 2021. **23**(1).
17. Yap, J., et al., *Role of Macrophages in Cardioprotection*. Int J Mol Sci, 2019. **20**(10).
18. Mouton, A.J., et al., *Mapping macrophage polarization over the myocardial infarction time continuum*. Basic Res Cardiol, 2018. **113**(4): p. 26.
19. Zaman, R., et al., *Selective loss of resident macrophage-derived insulin-like growth factor-1 abolishes adaptive cardiac growth to stress*. Immunity, 2021. **54**(9): p. 2057-2071 e6.
20. Sin, W.X., et al., *IRF-7 Mediates Type I IFN Responses in Endotoxin-Challenged Mice*. Front Immunol, 2020. **11**: p. 640.
21. Simons, K.H., et al., *IRF3 and IRF7 mediate neovascularization via inflammatory cytokines*. J Cell Mol Med, 2019. **23**(6): p. 3888-3896.
22. Wu, M., et al., *Interferon regulatory factor 7 (IRF7) represents a link between inflammation and fibrosis in the pathogenesis of systemic sclerosis*. Ann Rheum Dis, 2019. **78**(11): p. 1583-1591.
23. Burckhardt, C.J., J.D. Minna, and G. Danuser, *Co-immunoprecipitation and semi-quantitative immunoblotting for the analysis of protein-protein interactions*. STAR Protoc, 2021. **2**(3): p. 100644.
24. Soond, S.M. and A. Chantry, *Selective targeting of activating and inhibitory Smads by distinct WWP2 ubiquitin ligase isoforms differentially modulates TGFbeta signalling and EMT*. Oncogene, 2011. **30**(21): p. 2451-62.
25. Maddika, S., et al., *WWP2 is an E3 ubiquitin ligase for PTEN*. Nat Cell Biol, 2011. **13**(6): p. 728-33.
26. Caillaud, A., et al., *Regulatory serine residues mediate phosphorylation-dependent and phosphorylation-independent activation of interferon regulatory factor 7*. J Biol Chem, 2005. **280**(18): p. 17671-7.
27. Schmid, S., D. Sachs, and B.R. tenOever, *Mitogen-activated protein kinase-mediated licensing of interferon regulatory factor 3/7 reinforces the cell response to virus*. J Biol Chem, 2014. **289**(1): p. 299-311.
28. Maekawa, M. and S. Higashiyama, *KCTD10 Biology: An Adaptor for the Ubiquitin E3 Complex Meets Multiple Substrates: Emerging Divergent Roles of the cullin-3/KCTD10 E3 Ubiquitin Ligase Complex in Various Cell Lines*. Bioessays, 2020. **42**(8): p. e1900256.
29. Metzger, M.B., et al., *RING-type E3 ligases: master manipulators of E2 ubiquitin-conjugating enzymes and ubiquitination*. Biochim Biophys Acta, 2014. **1843**(1): p. 47-60.
30. Huang, X., et al., *The many substrates and functions of NEDD4-1*. Cell Death Dis, 2019. **10**(12): p. 904.

REVIEWERS' COMMENTS

Reviewer #1 (Remarks to the Author):

Minor comments:

- It is unclear why there are less values for LVMI than for LVID - please explain
- Fig S16 b) - please describe what indicates each dot on cardiomyocyte size plot and how it was calculated
- I suggest to move cardiomyopathy data from Fig S16 to Fig4 in order readers have better overview on the phenotype
- Fig S15 - please describe C- and N-terminal and FL region on the scheme a). Please describe also in the figure legend primer design for b) (i.e. which exons they amplify)

Reviewer #2 (Remarks to the Author):

I appreciate the authors thoroughness and careful consideration of reviewer concerns. I agree that the major questions have been addressed and additional questions outside of the scope of this manuscript may need to be addressed in subsequent studies.

Reviewer #3 (Remarks to the Author):

The authors have provided explanations and newly generated data to address the previously raised issues by this reviewer. Given the vast amount of data already presented in this paper, I agree that some of the concerns should be experimentally addressed in future directions.

Point by point rebuttals

- *It is unclear why there are less values for LVMI than for LVID - please explain.*

Answer: In the revised manuscript, we have now included all data points in the LVMI panel, which match the number of data points used for LVID.

- *Fig S16 b) - please describe what indicates each dot on cardiomyocyte size plot and how it was calculated*

Answer: We now detail that to calculate the cross-section area of cardiomyocyte we used Rhodamine Wheat Germ Agglutinin (WGA, Vector laboratories, #RL-1022) to stain the myocytes. Cardiomyocyte cross-sectional area (CSA) was determined on digitized images of WGA-staining; such images were obtained using a fluorescence microscope (Olympus BX61) attached to a digital camera and connected to a computer equipped with image analysis software. Cardiomyocytes' outlines were traced and cell areas measured with Image-Pro Plus 6.0 (Media Cybernetics, Inc., Silver Spring, Md). One view including at least 80 measurable cross sections of myocytes were measured and the average was calculated. The mean cardiomyocyte CSA of 4 views from each mouse was indicated as one dot in the plot. The detailed methods have been added in the revision, and in the legend for **Supplementary Figure 16b**.

- *I suggest to move cardiomyopathy data from Fig S16 to Fig4 in order readers have better overview on the phenotype*

Answer: Our study is about how WWP2 regulates immune cardiac cells in fibrosis. We present the ancillary cardiomyopathy phenotype data in full as supplementary data.

- *Fig S15 - please describe C- and N-terminal and FL region on the scheme a). Please describe also in the figure legend primer design for b) (i.e. which exons they amplify)*

Answer: Three main WWP2 gene isoforms have been previously described [1] and contain different protein domains: (1) a full-length isoform (WWP2-FL, covering the entire gene), (2) a N-terminal isoform (WWP2-N, capturing the 3' end of the protein) and (3) a C-terminal isoform (WWP2-C, containing the 5' end of the protein). To assess the *Wwp2* mRNA expression, we used primers targeting exon 9 of the WWP2-FL. This has now been specified in the legend for **Supplementary Figure 15**.

References

1. Chen, H., et al., *WWP2 regulates pathological cardiac fibrosis by modulating SMAD2 signaling*. Nat Commun, 2019. **10**(1): p. 3616.